

# Hydrometeorological drivers of the 2017 flood in the Brahmaputra basin in Bangladesh

Sazzad Hossain[1,2], Hannah L. Cloke[1,3,4,5], Andrea Ficchì[1], Andrew G. Turner[3,6], Elisabeth M. Stephens[1]

[1]Department of Geography and Environmental Science, University of Reading, Reading, UK
[2]Flood Forecasting and Warning Centre, BWDB, Dhaka, Bangladesh
[3]Department of Meteorology, University of Reading, Reading, UK
[4]Department of Earth Sciences, Uppsala University, Uppsala, Sweden
[5]Centre of Natural Hazards and Disaster Science, CNDS, Uppsala, Sweden
[6]National Centre for Atmospheric Science, University of Reading, Reading, UK

*Correspondence to:* Sazzad Hossain (mdsazzad.hossain@pgr.reading.ac.uk) and Elisabeth M. Stephens (elisabeth.stephens@reading.ac.uk)

**Abstract.** Flooding is a frequent natural hazard in the Brahmaputra basin during the South Asian summer monsoon. Understanding the causes of flood severity is essential for flood management decisions, but to date there has been little attempt to identify sub-seasonal variability of flood characteristics and drivers for the Brahmaputra in Bangladesh. In the 2017 summer monsoon, there was severe flooding in Bangladesh, but the Brahmaputra River, as well as its tributaries, behaved unusually compared to previous major flood events. This study analyses different hydrometeorological drivers of these floods, providing valuable information for the assessment and forecasting of future flood events. Water level and river flow time series have been decomposed using wavelet analysis to study the temporal variability within the hydrological cycle. During the 2017 monsoon, the extreme rainfall in August caused the water level of the Brahmaputra river and its tributaries to rise rapidly and exceed their previous historical record. This heavy rainfall was associated with a northward shift of the monsoon trough, creating active monsoon conditions in the Brahmaputra basin. The rainfall was localised over the lower sub-basins adjacent to the northern border of Bangladesh. The estimated river discharge in 2017 was slightly lower than the two previous major flood events in 1998 and 1988. The wavelet analysis of both daily water level and discharge shows that a high frequency component drove the severe flooding in 2017, compared to the low frequency component in 1998, where widespread basin accumulated rainfall acted as main driver of the flooding. The study concludes that the location and magnitude of extreme rainfall are key drivers controlling on the characteristics of the Brahmaputra floods. Understanding these drivers is essential for flood forecasting, in order to predict the timing, magnitude and duration of flooding, and also for understanding future climate change impacts on flooding. The study recommendations include analysing the synoptic situation along with different intra-seasonal oscillations as well as considering the spatial location of rainfall events for flood forecasting.



## 1 Introduction

The Brahmaputra is one of the most flood-prone basins in Bangladesh where flooding occurs annually during the
monsoon season (June to September). The monsoon onset initially takes place over the Meghna and Brahmaputra
basins, while later it moves north-westwards towards the Ganges basin (Fig. 1). Flooding in Bangladesh is defined
by the Flood Forecasting and Warning Centre (FFWC) in terms of a 'danger level' at which water starts to cause
damage to property, crops or other infrastructure; a river is said to be in a 'flooding situation' when water levels
cross the danger level, and in a 'severe flooding' situation when water levels exceed the danger level by 1 m or
more. The danger level (DL) is defined either based on the design level of the flood protection embankments or
the average flood (2.33-year return period) of a flood-monitoring gauging station (BWDB, 2004). During the 2017
monsoon two successive floods hit the Brahmaputra basin during July and August, and the water level of the
Brahmaputra and its tributaries the Teesta and Dharla rose rapidly to exceed their previous historical records,
reaching more than 1 m above their respective flood danger levels. The Brahmaputra, Teesta, Dharla and
Dudkumar exceeded their flood danger level 1 to 3 day after the water level started to rise (2 day for Dudkumar
and Dharla,1 day for Teesta and 3 day Brahmaputra). The flooding caused severe damage to crops as well as
physical infrastructure. Around 8 million people were affected by the flood, in which 0.6 million hectares of crop
lands were damaged and 150 people died (DDM, 2017). Similar flooding was also experienced upstream of
Bangladesh.

The meteorological forcings that cause heavy rainfall in the Brahmaputra basin during the summer monsoon are
the movement of the eastern end of the monsoon trough to the Assam region, producing break monsoon conditions
over central India and active conditions around the Himalayan foothills, often with a monsoon depression originating
from the Bay of Bengal and cyclonic circulation over the basin (Dhar and Nandargi, 2003; Dhar and Nandargi,
2000; Nandargi and Dhar, 2011). Sub-seasonal variation of monsoon rainfall is marked by wet and dry spells known
as active and break events, with typical lifespans of around 2 weeks (Krishnamurthy and Shukla, 2007). Active and
break events form part of the 30-50 day intraseasonal variation known as the Boreal Summer Intraseasonal
Oscillations (BSISO), featuring northward-propagating bands of convection at South Asian longitudes together with
eastward propagation at the equator, akin to the Madden Julian Oscillation (MJO) (Annamalai and Sperber, 2005).
Meanwhile at interannual time scales, the El Niño Southern Oscillation (ENSO) is the most well-known mode of the
global circulation understood as a driver of variability of monsoon rainfall (Goswami et al., 1999; Krishnamurthy and
Kinter, 2003). There is a general tendency for El Niño and La Niña to correspond to dry and wet conditions
(respectively) over the Indian region. However, in reality the relationship is more complex and some droughts and
flood events are found not to occur during El Niño and La Niña years (Emerton et al., 2017; Goswami, 2005).



The 2017 water level rise (cm/day) and magnitude (m) of the Brahmaputra were exceptional compared to other years. On the other hand, the river flow in 2017 was relatively lower than previous severe flooding years 1988 and 1998 (see Figure 11 and Table S3). Understanding the major drivers for this flooding in comparison to previous flood events is key to informing the development of reliable early warning systems and accurate predictions of future flood hazard in a changing climate. Therefore, the aim of this study is to consider the meteorological and hydrological drivers which were responsible for the 2017 monsoon flooding in the Brahmaputra basin in comparison with previous years.

Our objectives are to address the hydrometeorological drivers of the flooding by analysing antecedent conditions (Blöschl et al., 2013; Schröter et al., 2015), large-scale atmospheric and ocean anomalies (Paeth et al., 2011), the synoptic situation (Blöschl et al., 2013) and extreme statistics of precipitation and river flow (Schröter et al., 2015). Several studies, e.g. Islam and Chowdhury (2002), Mirza (2003), Islam et al. (2010) have been carried out to assess the hydrological and meteorological characteristics of major flood events within the main river basins in Bangladesh. However, previous studies have evaluated monthly rainfall during monsoon months and are therefore unable to examine the nature of high frequency rainfall events inherent in daily data that trigger flooding.

The paper is structured as follows: A short description about the basin is provided in Section 2. Sections 3 and 4 describe the data and methods respectively. The results are presented in Section 5. Finally, discussion and conclusions are summarised in Sections 6 and 7 respectively.

## 2 Characteristics of the Brahmaputra basin

The Brahmaputra basin is located between approximately 83°E-96.5°E and 24.5°N-30.25°N with a total area of 580,000 km². It is a trans-boundary river basin shared by Bangladesh (8.1%), Bhutan (7.8%), China (50.5%) and India (33.6%) (Goswami and Das, 2002). It originates from glaciers in the Kailash range in Tibet at an elevation of about 5300 m and flows through China, Arunachal Pradesh and Assam states of India before finally meeting with the Ganges river in Bangladesh (Goswami and Das, 2002). The river length is about 2880 km of which 1625 km is in Tibet (China), 918 km in India and 337 km in Bangladesh (Sarma, 2005). After meeting with the Ganges, it is named the Padma and finally meeting with the Meghna river at Chandpur before flowing into the Bay of Bengal (Fig. 2). The physiography of the basin can be classified into three distinct zones: the Tibetan plateau (elevation exceeding 3500 m); the Himalaya belt (elevation between 100 m and 3500 m) and the agricultural flood plain (elevation up to 100 m) (Immerzeel, 2008). The lower flood plain (Assam and Bangladesh) of the basin is flood prone and affected by monsoon flooding almost every year.



Approximately 60-70 % of the annual rainfall occurs in the monsoon period from June to September (Immerzeel, 2008). The Arunachal Pradesh, Assam and sub-Himalayan regions (upstream of Bangladesh) form the high precipitation zone in the basin with mean annual rainfall of about 2300 mm. Some places in the foothills of the Himalaya receive rainfall as high as 5000 mm in a year (Dhar and Nandargi, 2000; Singh et al., 2013). On the other

hand, the high altitude part of the basin in the Tibet region has an average yearly rainfall of 734 mm (Immerzeel, 2008). Glacial snowmelt in the Tibet area provides flow in the upstream part of the Brahmaputra river (Immerzeel, 2008; Masood and Takeuchi, 2015; Paura, 2004). The maximum discharge recorded is 102,535 m$^3$/s at Bahadurabad stream gauging station located in the flood plain river delta in Bangladesh on 9th September 1998 (see Fig. 2 for location). The present study focuses on the lower part of the basin located in Bangladesh (Fig. 2).

## 3 Data

### 3.1 Hydrological data

The study uses daily water level and river flow to analyse the hydrological characteristics of the 2017 flooding. The data has been obtained from the hydrological division of the Bangladesh Water Development Board (BWDB). The four representative flood monitoring water-level gauge stations – Bahadurabad, Dalia, Kurigram and Pateswari (Fig. 2) on the Brahmaputra, Teesta, Dharla and Dudkumar rivers respectively – have been used for the 31-year study period 1987 to 2017, which contains severe, normal and no-flooding years. At Bahadurabad, river flow is

measured using a current meter (or Acoustic Doppler Current profiler) approximately twice a month to compare with water-level data collected using a manual water-level staff gauge at 3-hour intervals five times per day between 6:00 AM and 6:00 PM local time, with no data at night (see Fig. S1 for water-level gauge and river flow measurements). While the correlation between water level and measured discharge is reasonably high (r$^2$=0.93) at Bahadurabad (Fig. S2a), there is still considerable uncertainty in the estimation of daily river flow especially for

high flows (Fig. S2b).

### 3.2 Meteorological data

### 3.2.1 Observed rainfall

The observed rainfall data has been used to study the rainfall extreme basin located inside Bangladesh and data has been collected from the Bangladesh Meteorological Department (BMD) for the period 1987 to 2017. There are six rain gauge stations located inside the Brahmaputra basin in Bangladesh (Fig. 2). Rainfall data is collected manually by the BMD using tipping-bucket rain gauges at 3-hour intervals, and daily accumulated rainfall is

calculated from these data. For the present study, daily accumulated rainfall data has been used. The





APHRODRITE gridded precipitation dataset for the period 1951 to 2007 has also been used to study mean monsoon rainfall and active / break rainfall anomalies (Yatagai et al., 2012).

### 3.2.2 TRMM rainfall data

The Brahmaputra is a transboundary river basin and observed rainfall data is not available from the upstream basin, therefore TRMM (Tropical Rainfall Measurement Mission) daily data has been used to investigate rainfall conditions in the wider basin. The TRMM 3B42 daily data product version 7 is obtained from the Goddard Earth Sciences Data and Information Services Center (DISC, 2016), consisting of TRMM as well as other satellite estimates of

precipitation known as TRMM Multi-Satellite Precipitation Analysis (TMPA). This product has a spatial resolution of 0.25° and covers 50°S to 50°N in latitude (Liu et al., 2012). Raw TRMM data is 3-hourly, thus daily data (mm/day) is obtained by summing 3-hour accumulated precipitation for each day (Scheel et al., 2011). Zulkafli et al. (2014) have compared TRMM 3B42 versions 6 and 7 over other tropical mountainous areas in the northern Peruvian Andes and found considerable improvements especially due to new ground clutter algorithms in version 7.

### 3.2.3 Other meteorological and climate data

Besides the use of rainfall data, other meteorological variables including daily mean sea-level pressure, 850 hPa geopotential height, wind vectors and precipitable water have been obtained from the NCEP reanalysis data (Kalnay

et al., 1996) in order to study the synoptic situation for the two major rainfall events of the 2017 monsoon season (1 July to 7 July and 8 August to 13 August 2017). Other large-scale climate data were used to define ENSO, Madden-Julian Oscillation (MJO) and Boreal Summer Intraseasonal Oscillation (BSISO) conditions. For ENSO, the monthly Oceanic Niño Index (ONI) index time series in NINO3.4 region was obtained from (NOAA, 2019) based on the monthly global analysis Extended Reconstructed Sea Surface Temperature (Huang et al., 2017). El Niño and La Niña years

were selected by applying a tercile-based approach on the ONI index from August to December. MJO conditions have been studied using phase diagram available from BOM (2017) and BSISO conditions have been studied using principle components (PC1 and PC2) of the BSISO mode (Kikuchi et al., 2012).

### 4 Methodology

### 4.1 Wavelet analysis of hydrological time series

In this study, the 1-D discrete wavelet transform (DWT) has been used to remove noise from Brahmaputra river water level and discharge data to identify short-term variations and the annual cycle and allow comparison between different

years. This approach gives an indication of the relative importance of the role of seasonal hydrological cycle compared to specific monsoon rainfall events for different flooding years. Wavelet analysis is used to decompose a time series



into time-frequency space and commonly used to study the behaviour of the local variation of a time series (Torrence and Compo, 1998). The Daubechies wavelet function is commonly used in hydrometeorological time series analysis (Center, 2016; Chen et al., 2016; Franco Villoria et al., 2012; Kumar et al., 2017; Zhang et al., 2017). The Daubechies wavelet function has been used here to decompose the daily water level of the Brahmaputra from 1987 to 2017 to

obtain high and low frequency components. The daily time series is decomposed into 6 detailed components (D1-D6) and one approximation (A6). The detailed components present the variation in $2^n$ (dyadic translation) where n is the level of the detailed component. The daily time series D1 to D6 therefore represent 2-day, 4-day, 8-day, 16-day, 32-day and 64-day periodicity respectively. For more detailed information on this method, the reader is referred to the study Zhang et al. (2017).

## 4.2 Flood characteristics and extreme value analysis of hydrological data

Hydrological characteristics have been studied in terms of the rate of rise of the water level, flooding duration and timing and maximum water level and discharge for different flooding and non-flooding years. High frequency (sub-

seasonal variation) and low frequency (general trend) components of the wavelet analysis of water level and discharge are compared with other flooding and non-flooding years to study how these components are different in the 2017 flooding. The exceedance probability of annual maximum discharge of the Brahmaputra river has been calculated using the Generalized Extreme Value (GEV) distribution for both river flow and water level; the parameter values (scale, shape and location) are estimated using the Maximum Likelihood method. This allows a comparison

of the 2017 flood in comparison to other years.

## 4.3 Meteorological characteristics and extreme value analysis of hydrological data

Meteorological characteristics have been analysed in terms of the synoptic situation during the heavy rainfall

events. The rainfall anomalies are presented for the months of April to September using TRMM daily rainfall data to find the departure from the climatological mean rainfall. The basin cumulative rainfall for the monsoon months (June to September) has been calculated based on the basin area for both the flooding and non-flooding years from 1998 to 2017 to describe its interannual variability. The FFWC identifies 1998, 2004, 2007, 2012 as severe flooding years (FFWC, 2016). The rainfall distributions of flood-triggering rainfall events are studied for their location

of occurrence. A depth-duration-frequency (DDF) curve has been developed to analyse the rainfall extremes for the 2017 season at six gauge stations within the basin in Bangladesh. The frequency analysis is performed using the Generalized Extreme Value (GEV) distribution of the maximum rainfall for 1 to 10-day duration to construct DDF curves.



## 5 Hydrometeorological analysis of the 2017 flood

### 5.1 Meteorological characteristics

After the monsoon onset, the rainfall during June to September consists of wet and dry spells in the basin. Figures 3a and 3b show basin-average rainfall (mm/day) and daily rainfall recorded by a gauge station respectively, whereas Fig. 3c shows the discharge hydrograph of the Brahmaputra river at Bahadurabad during June to September in 2017. The basin-average rainfall and the gauge data clearly show several distinct rainfall events during the monsoon period. Figure 4a shows the distribution of climatological mean daily rainfall over the South

Asian region during June to September for 1951 to 2007. The mean monsoon rainfall shows the lower part of the Brahmaputra basin to be wetter than the upper part (Fig. 4a) and July is the wettest month (Fig. 4b). The heavy rainfall events in the Brahmaputra basin are associated with break events in central India and active events in the Himalayan foothills. The box in Fig. 4(a,c and d) represents the 'core monsoon zone' (Dunning et al., 2015), which is representative of intraseasonal monsoon variability from an Indian perspective. The active phase is defined by

the standardized rainfall anomaly exceeding +1 in the monsoon core zone, while a break phase is defined if the standardized rainfall anomaly is below -1 for at least three consecutive days during July and August (Rajeevan et al., 2010). Figures 4c and 4d show composites of active and break rainfall anomalies prepared using the Aphrodite gridded precipitation data over the 1951-2007 period. During (Indian) break monsoon conditions the monsoon trough shifts northwards close to the foothills of the Himalaya compared to its active phase, as determined both in

the 850 hPa geopotential height field and surface charts (Dhar and Nandargi, 2003; Rajeevan et al., 2010). As clearly seen in Fig. 4d, break conditions over central India are commonly associated with enhanced rainfall over the Brahmaputra basin. The mean daily composites of 850 hPa geopotential height, indicating the location of the monsoon trough during active and break events, are presented in Fig. 4e and 4f respectively. These clearly indicate that during monsoon breaks over central India, the eastern end of the monsoon trough becomes situated over the

Brahmaputra basin (Fig. 4f), which would draw moist southerly winds into Bangladesh from the Bay of Bengal. During active conditions on the other hand (Fig. 4e), the lower tropospheric circulation would turn cyclonically, leading to south-easterly wind anomalies which return the airflow to northern India.

Next, we describe the meteorological characteristics associated with the 2017 flood in the Brahmaputra basin, beginning with an overview of the synoptic situation.

### 5.1.1 Synoptic situation

The synoptic features of the two rainfall events during 1-7 July and 8-13 August 2017 which were associated with the flooding events in the Brahmaputra, revealed that the monsoon trough shifted towards the foothills of the

Himalaya and the Assam region. This was associated with break conditions over central India and rainfall in the



foothills of the Himalaya and the Assam region. The shifting of the monsoon trough is captured by the negative geopotential height anomaly at 850 hPa in the Himalayan foothills and surrounding region (Fig. 5a-b), somewhat consistent with the composite anomalies in several Indian monsoon break events in Fig. 4f. Similarly, the mean sea-level pressure also shows negative anomalies in the Brahmaputra basin area (Fig. 5 c-d). The wind vectors

show south-westerly flow into the Brahmaputra basin from the Bay of Bengal, while vertically integrated moisture fluxes show strong moist flows from the Arabian Sea, across the Indian subcontinent and again over the Bay of Bengal before reaching Bangladesh (see Fig. S3). The anomaly field of precipitable water (Fig. 5e-f) shows additional available moisture over the lower sub-basin (Assam, sub-Himalaya region) during the two rainfall events. Due to the shifted position of the monsoon trough, a strong pressure gradient exists, leading to the moisture-laden

air mass being transported into the Brahmaputra basin area and leading to the heavy rainfall events in July and August. The daily weather bulletin of Indian Meteorological Department (IMD) also shows the monsoon active conditions that prevailed during 8-13 August in this region (Table S1). Due to the shifting of the monsoon trough, during this time the lower part of the basin located both in Bangladesh and Assam received heavy rainfall (e.g. 545 mm of rainfall over 5 day recorded at a gauge station in Bangladesh).

### 5.1.2 Large-scale climate drivers

In order to determine whether large-scale modes of climate variability had any bearing on the Bangladesh flooding of 2017, we examine ENSO conditions. First, we consider a composite difference between El Niño and La Niña

events for rainfall in the Brahmaputra basin. Figure 6 (a) shows the difference of June to September rainfall between La Niña and El Niño years. The difference indicates that La Niña years have relatively more rainfall than El Niño for the Brahmaputra basin. ENSO conditions for 2017 are presented using an Oceanic Niño Index (ONI) (Fig. 6b). ONI is defined as the 3-month running-mean SST anomaly in the Niño-3.4 region (5°N–5°S, 120°–170°W). An El Niño state is defined when ONI exceeds 0.5°C, while La Niña is characterized by an ONI below -0.5°C, in each

case persisting for at least 5 months (Kousky and Higgins, 2007). The ONI index indicates that 2017 was somewhere between neutral and weak La Niña conditions (Fig. 6b).

Now to examine if modes of intraseasonal variability were present which may have forced the heavy rainfall events over the Brahmaputra basin, we examine the behaviour of the Madden-Julian Oscillation (MJO) and Boreal Summer Intraseasonal Oscillation (BSISO) in 2017. The MJO when under phases 7, 8, 1 and 2 shows Indian

monsoon break conditions (and therefore active conditions over the Brahmaputra basin) whereas phases 3, 6 and 4 shows the opposite (Pai et al., 2011). The different MJO phases were studied using the Wheeler-Hendon phase diagram based on Real-time Multivariate indices (RMM1 and RMM2) for the 2017 monsoon months (Wheeler and Hendon, 2004). In June 2017, the MJO was active in phase 3, while it was in a weak state for the remaining monsoon months (July-September; see Fig. S4). Therefore it appears that the MJO had no bearing on monsoon





variability in Bangladesh during 2017, which suggests little impact of this eastward propagating, equatorial phenomenon. Finally, we examine indices of the BSISO to find the influence of northward propagation. The BSISO mode shows strong northward propagation between the end of July to mid-August during the 2017 monsoon (Fig. 6c), suggesting it may have played a role in the synoptic activity leading to the 2017 Brahmaputra flooding.

### 5.1.3 Monthly rainfall totals

While April 2017 shows generally positive rainfall anomalies across most parts of the basin, the first two months (June-July) of the monsoon season were drier than normal (Fig. 7c-d). However, the rainfall anomalies in August 2017 show strong anomalies above normal rainfall in the lower part of the basin especially in Bangladesh and adjacent areas (Fig. 7e). This is in contrast to 1998, where the rainfall in the premonsoon period (April-May) was not significant (Fig. 7g-h), but June to August had basin-wide positive anomalies (Fig. 7 i-j-k). Next, Fig. 8 shows the cumulative rainfall from June to September for the basin. The cumulative rainfall was not particularly high at the beginning of the monsoon and was less than other flooding years. From the end of June until July, rainfall was similar to 1998. However, unlike 1998, where the rate of rainfall accumulation was quite steady, in 2017 there were more fluctuations with a sudden jump visible in the curve in early August. Such an abrupt step-increase in rainfall rates is likely to lead to an important infiltration-excess runoff component and a sudden influx to the river. The year 1998 featured a rather linear increase in accumulated rainfall, whereas 2017 shows a clear large and rapid increase in rainfall accumulation between the end of July and around 10 August, and also given the already high-inflow conditions at that time of the year flooding would be likely. The other noticeable year with such a rapid period of accumulated rainfall increase is 2004 (purple) in early July. However, the final basin accumulated rainfall total for 2017 was higher compared to other years (Fig. 8). Figure 8 suggests that rainfall at the beginning of the monsoon was not particularly important in contributing to the severity of the 2017 flooding.

### 5.1.4 The characteristics of the 2017 flood-generating rainfall events

Two rainy events were found to control the flood peak and timing in the 2017 monsoon (Fig. 3). The rainfall events during the monsoon are presented by the gauge data (see Fig. 3b), and the spatial patterns of these rainfall events are presented in Fig. 9. The amount of rainfall during the July rainfall event was less than that of the August event, as reflected in the water levels and river flow in the corresponding flood event. The second rainfall event was more intense and occurred between 8-13 August in northwest Bangladesh and nearby upstream sub-basins (Assam, Arunachal Pradesh, Bhutan) and the sub-Himalayan region of West Bengal (Jalpaiguri, Coochbehar, Alipurduar) (Fig. 9b). Between 8-13 August, the Syedpur gauge station recorded 1-, 2-, 3-, 4- and 5-day accumulated rainfall totals of 287 mm, 455 mm, 493 mm, 528 mm and 545 mm respectively. Similar heavy rainfall was recorded at


Indian rain gauges adjacent to the Bangladesh border (one day maximum rainfall recorded at Domohani 250 mm, Alipurduar 390 mm and Jalpaiguri 290 mm on 12 August; see Fig. 2b for station locations) (IMD, 2017b). The synoptic situation shows that during this period the monsoon was active over this area of the basin, coinciding with the rainfall events listed.

5 Point data from six rain-gauge stations inside Bangladesh are analysed to evaluate rainfall extremes during June to September 2017. Syedpur and Rangpur are located upstream of the Bahadurabad river gauging station whereas Bogra, Tangail, Mymensingh and Dhaka are located further downstream (see Fig. 2 for locations). The upstream rain gauges receive more rainfall than the downstream gauges, but only rainfall of 2-day and 4-day duration in Syedpur exceeded the previous records (Fig. 10a). The rainfall at the other stations did not exceed historical records for the 10 same duration and was much lower for all durations, which indicates that the heavy rainfall was localized. There are variations in the return periods of rainfall, estimated by GEV fit for different durations and for different gauges (Fig. 10b). For the Syedpur station, rainfall was at its most extreme for the 5-day maximum rainfall with an annual exceedance probability of 2.5% (40-year return period).

15 **5.2 Hydrological characteristics of the 2017 flood**

During the monsoon the Brahmaputra river experiences wave-like fluctuations in response to rainfall pulses over the basin, as demonstrated by its hydrograph (Fig. 11a), though not all the peaks do not cross the flood danger level. The hydrological regime has a distinct annual cycle with the peak water level/river flow between July and 20 August (Fig. 11b and 11e). The flood water usually starts to recede from the end of August, however in some years flooding continues until the second week of September. The flooding in 2017 shows distinct characteristics in terms of its highest water level; the river flooding occurred between 7-17 July and 11-24 August, but in August the flood peak was more severe, and the water level exceeded the previous historical peak (Fig. 11a). Both the water level and river flow hydrograph of the Brahmaputra river have been decomposed into high frequency (variability within 25 the hydrological cycle) and low frequency components (residual hydrograph when the high frequency components have been removed, i.e. seasonal regime) to investigate the variability during the 2017 monsoon flood compared to other years. While the highest-ever water level was reached in 2017 (Fig. 11a), the smoothed curve of the seasonal regime (general trend, A6) was not exceptional (Fig. 11b). The river water level of the Brahmaputra has a clear high frequency (D4/16-day) component (Fig. 11c), showing strong sub-seasonal variability. We analyse 30 here the 16-day periodicity (D4) component of the Daubechies wavelet function (Fig. 11c and 11f), since it resulted to be the dominant high-frequency component, which is lined with the wet and dry spell typical time scales. During the 2017 flood, the high frequency component reached an exceptionally large amplitude and corresponds well to the recorded water level in August (Fig. 11c). The wavelet analysis of river flow also shows similar results to the



water-level analysis (Fig. 11e and 11f). In comparison to other severe flooding years, 1998 presented a lower short-term variability (Fig. 11c and 11f), but the trend component (seasonal regime) was higher than all other years for both river flow and water level (Fig. 11b and 11e).

### 5.2.1 Rate of water-level rise

The rate of water level rise is important in order to forecast and provide timely flood warnings, as it determines how quickly the water level will cross the flood danger level and how quickly flood managers and communities need to take action. The Brahmaputra river at the Bahadurabad station shows a high but not record-breaking rise of water level during the 2017 flood (Fig. 12a). The mean water-level rise of the tributaries was similar to other years (Fig. 12b, 12c and 12d). The higher rate of rise in the Brahmaputra river was likely due to the concurrent contributions from its tributaries, due to rainfall over adjacent areas in the lower sub-basins forming a hydrological sweet spot.

### 5.2.2 Flood duration

The flood duration on the main Brahmaputra river and the Teesta, Dharla and Dudkumar tributaries in 2017 was not particularly remarkable, especially compared to the exceptional flood duration in 1998 (Fig. 13). The flood duration in 1998 was extraordinarily long (66 days above danger level at Bahadurabad gauge station), while during the 2017 monsoon, the flood duration of the Brahmaputra river was 25 days, which was higher than 2004 and 2007 and similar to 1988 (Fig. 13a). Flood durations in other years are found to vary (typically) from 2 to 20 days. Flooding usually has a shorter duration in the tributaries (Teesta, Dharla and Dudkumar) than that of the main course of the Brahmaputra river (Fig. 13b, 13c and 13d).

### 5.2.3 Peak water level

The peak water level was the most striking feature in the 2017 flood as the Brahmaputra, the Dharla and the Teesta exceeded their previously recorded highest water level (Fig. 14a). The previous highest water level of the Brahmaputra at Bahadurabad was above the danger level by 112 cm and 121 cm in the 1988 and 2016 monsoon seasons respectively. The peak water level of the Brahmaputra, during 2017 was 134 cm above danger level and the estimated exceedance probability of peak water level was 1.3%. The peak water levels of the Dharla, Dudkumar and Teesta were 134, 137 and 65 cm above danger level respectively, with the Dudkumar slightly less than the previous recorded peak level (147 cm). In 2017 the Brahmaputra crossed the danger level twice (7 July and 11 August), a behaviour that is not unusual compared with other years (Fig. 14b). The usual date of exceeding the danger level is between July and August, though sometimes flooding also occurs in September before the withdrawal of the monsoon. The water level in 1998 flowed continuously above the danger level for two months



and the behaviour of flood was different from other years. Peak water levels of the Brahmaputra river and its three tributaries are provided in Table S2, with corresponding dates of occurrence.

### 5.2.4 Peak river discharge

When the measured discharge is not available on the day when the water level is maximum in the river, it is estimated using a rating curve. The measured discharge of the Brahmaputra river at Bahadurabad gauging station was 78525 $m^3$/s on 17 August 2017 with a water level of 20.79 m, while the peak water level was 20.84 m on 16 August with estimated river discharge from the rating curve of 93359 $m^3$/s. The previous measured and estimated

maximum discharges were 102535 $m^3$/s and 103128 $m^3$/s in 1998. The peak river discharge during the flood in 2017 has an estimated exceedance probability of 4.8% (Fig. 15), approximately four times as likely as the water level exceedance probability which was higher than all previous years (see Table S3 for maximum recorded river flow in different years). The estimated discharge in 2017 was less than that of both the 1998 and 1988 floods (Fig. 15).

### 6 Discussion

#### 6.1 Inter-annual variability in flood characteristics of the Brahmaputra basin

There is a significant year-to-year variability in flood magnitude, timing and duration, as well as in the spatial occurrence of flooding along the river network, in the Brahmaputra basin and several hydrometeorological drivers play an important role in shaping these flood characteristics. In this study, we have analysed different flood characteristics using hydro-meteorological data over a 31-year period, to understand what distinct features characterize flood behaviour across different flooding years and their link with multiple interdependent drivers of

flooding. Here below we discuss these characteristics.

**Flood duration:** The flood duration (number of days with water level above danger level) of the Brahmaputra river varies from minimum 2 days to maximum 66 days in the 31-year period (1987-2017), which indicates a very high inter-annual variability in flood duration and dynamics. The flood duration was exceptionally long in 1998 (66 days), while it was between 10 and 30 days for most other high-impact flooding years (including 1988, 2004, 2007, 2012

and 2017). The flood duration of the three tributaries, Dharla, Dudkumar and Teesta, is usually much shorter compared to the main river of the Brahmaputra (and most often below 16-20 days).

**Flood magnitude:** In our study, we found 7 years (1990, 1992, 1994, 2001, 2005, 2006 and 2009) out of 31 years when river flow remained below the flood danger level. Exceptionally high water level (water level more than 1 metre above danger level) was found in 1988, 2012, 2016 and 2017 flooding years. On the other hand, in some

other high-impact flooding years, such as 1998, 2004 and 2007, the peak water level was less than 1 metre above





danger level. Therefore, flood duration is one of the most important factors that impact on flood damage, as it may lead to greater damage even for lower flood magnitudes, as in 1998 (Hofer and Messerli, 2006)

**Flood timing**: There is significant variation of the timing of flooding in the Brahmaputra basin within the monsoon season. The river water levels start to cross the danger level from the end of June with the influence of monsoon rainfall over the basin area (both in Bangladesh and outside). It was found that flood occurred (i.e. water level crossing the danger level) 8%, 58 %, 21 % and 13% times in June, July, August and September respectively; thus, severe floods are most frequent in July, but are also common in August and September

**Spatial occurrence of flooding across the river network:** The flooding of the Brahmaputra basin in Bangladesh appears to become more severe and longer due to peak flow synchronization of the other two major rivers in the Bengal region, i.e. Ganges and the Meghna. If flood waters of these three rivers flow above danger level simultaneously then flood duration becomes longer (Mirza, 2003).

The rainfall characteristics such as intensity, location/extent and duration played the most important role in shaping flood characteristics in the Brahmaputra basin. For instance, positive rainfall anomalies were basin wide during the 1998 flood, and consequently the flood duration was exceptionally long, exacerbated by peak flow synchronization of the Brahmaputra with the Ganges and Meghna, where the three rivers combine. Conversely, in 2017 the heavy rainfall was more localized in the downstream part of the three sub-basins and this led to a shorter flood duration across the whole basin and a more rapid rise of water levels in the Brahmaputra river than other years, given the synchronous contribution from upstream. In some years, flooding occurred only in the tributaries depending on the rainfall situation of the corresponding sub-basins, showing that the spatial variability of rainfall plays a key role as a driver of flooding in the basin.

## 6.2 Drivers of the 2017 flood and implications for climate change

The study separates high frequency and low frequency components of both water level and river flow data of the last 31 years using wavelet analysis and identifies the dominant short-term variation with a time scale of 16 days, which is linked with the monsoon rainfall temporal dynamics (i.e. succession of wet and dry spells). We found that the relative importance of the seasonal regime and the high-frequency variability in flood signal varies markedly across the 31 years, with 2017 presenting the highest short-term variability on record, and 1998 presenting the most important low-frequency component.

Indeed, the 2017 flooding was hydro-meteorologically different to the highest impact floods in the basin in 1998. Cumulative rainfall over the basin in 2017 showed a rapid increase in mid-August due to intense rainfall; therefore, the high frequency component to the flood wave was exceptional in terms of both water level and discharge. Hence, we determine that short duration (e.g. up to a week) intense rainfall located over the sub-basins of the three tributaries in the lower part of the basin in August was the most important driver in 2017. The rainfall intensity in





the first flood event during July was less and spread further over the basin, therefore the rise of river water level was steady and river flow remained below danger level for all tributaries. It is perhaps the location of the following extreme rainfall event that was the key factor in August 2017 flood, with all three tributaries experiencing a concurrent rapid rise in water levels that entered the main river channel at a similar time. In contrast, rainfall
anomalies were basin wide during the 1998 flood, and consequently the flood duration was exceptionally long, exacerbated by peak flow synchronization with the Ganges and Meghna where the three rivers combine.

Our results point to two drivers for floods on the Brahmaputra river: basin-wide precipitation anomalies over the whole monsoon season (such as 1998), or more localized intense rainfall falling in hydrological sweet spots (as in 2017). While the 1998 flood had the higher impact, the 2017 flood was more of a challenge for forecasters.
Typically, the water level in the Brahmaputra rises gradually, which provides time to comprehend when the danger level may be crossed and implement appropriate flood preparedness measures. However, our results for the 2017 event show the importance of skilfully forecasting the magnitude and location of precipitation during the active phase of the monsoon, given the unusually rapid rise of water level driven by such a particular event. Our results also point to the importance of investigating how these two drivers for floods on the Brahmaputra river will evolve
in the future under global warming scenarios. To answer this question, it is essential to know how climate change is affecting the monsoon rainfall and its spatio-temporal characteristics. The future seasonal mean South Asian total rainfall shows an increasing trend in different climate projections (Turner and Annamalai, 2012). However, the spatial distribution of monsoon rainfall at the regional scale is more difficult to predict for the current generation of general circulation models, with larger uncertainties in the sub-seasonal and synoptic-scale characteristics (e.g.,
monsoon depressions, location of the monsoon trough) (Turner and Annamalai, 2012). Thus, there is larger uncertainty in the projections of future intense rainfall events at the synoptic scale, such as those responsible for the 2017 flooding. Some studies have shown increasing flood risk for the Brahmaputra basin in climate projections (Mohammed et al., 2017; Zaman et al., 2017), but these projections are more related to only one of the two drivers, i.e. the basin-accumulated rainfall controlling the flooding in 1998. However, most previous studies lack a
quantification of uncertainty in climate change projections of monsoon rainfall and how this uncertainty propagates into the hydrological hazard projection. Furthermore, flood attribution studies should consider these different flood drivers and characterise separately these different trends for a sound assessment, similarly to what has been proposed by Merz et al. (2012). In the Brahmaputra basin, it seems essential to predict whether the intense localized rainfall events (as in 2017) will increase in frequency and duration in the future, in order to estimate the
changes in the flooding hazard type that is the most challenging to forecast and prepare for floods.



### 6.3 Recommendations for flood forecasting

Given the analysis presented in this paper, the following recommendations are suggested for flood forecasting in the Brahmaputra basin.

**Synoptic situation:** Currently in FFWC in Bangladesh the synoptic situation is not fully considered in flood forecasting. Our work showed that an analysis of the synoptic situation over the lower sub-basins (Assam, Meghalaya and sub-Himalaya West Bengal region) could play an important role in anticipating heavy rainfall events and their contribution to flooding.

**Tropical Intra-Seasonal Oscillation (ISO) and large-scale climate modes:** The Boreal Summer Intra-Seasonal Oscillation (BSISO) mode needs to be carefully monitored during the monsoon period in order to consider its prominent northward propagation carefully and the associated active and break rainfall events. It is also important to consider the influence of other teleconnections, such as ENSO and the Indian Ocean Dipole (IOD), on regional rainfall, which is likely to be important for flood forecasting at seasonal time scales during the South Asian monsoon season.

**Central India and the Brahmaputra basin and/Bangladesh rainfall dipole:** Our work demonstrated that a wider spatio-temporal analysis of current and future predicted monsoon rainfall events can be used to anticipate flood situations in the Brahmaputra basin. Less rainfall in central India can indicate more rainfall in the Brahmaputra basin.

**Hydrological sweet spots:** Our work has shown the importance of potential hydrological sweet spots in the lower sub-basins of the Brahmaputra (near the Bangladesh border), where heavy rainfall events contribute to a more rapid rise of river water levels. The location of the rainfall events may also contribute to determine the flood timing and magnitude, with a possible synergistic effect for increased flood hazard produced by a synchronised flood wave from the three tributaries. Therefore, the spatial distribution of rainfall at this sub-basin scale is essential for flood forecasting in the Brahmaputra.

**Rate of rise of water levels:** Another important consideration is the potential rate of rise of the water level in the river. Our work has shown that high rates of water level rise (e.g. 50 cm/day) can occur over several consecutive days and such behaviour has not been previously considered for the Brahmaputra flood forecasting.

**Ensemble forecast:** Operational implementation of ensemble river flow forecasting techniques for the Brahmaputra would allow a better appreciation of the upcoming flood hazard considering the uncertainties in the forecast and allowing better decision to be taken for flood preparedness (Cloke and Pappenberger, 2009; Demeritt et al., 2010; Nobert et al., 2010)





## 6.4 Looking Forward

The Brahmaputra is a braided river and morphological changes are common due to sedimentation. Further work could analyse the spatial extent of inundated areas and the role of river morphological changes of the Brahmaputra

and its tributaries, as a possible important driver of flood characteristics. Also, back water effects of the main river and its influence on the flooding of the tributaries needs further investigation.

A detailed study is also needed to investigate the influence of intra-seasonal modes such as the Madden-Julian Oscillation (MJO) and BSISO and their relationship with Brahmaputra basin floods in the long-term record. Also, a relationship between sub-seasonal variations of river flow and corresponding temporal variability of precipitation

events is recommended for further research.

Presently, ensemble hydrological forecasts from the Global Flood Awareness System (GloFAS) (Alfieri et al., 2013), are run by the ECMWF (European Centre for Medium-Range Weather Forecasts) as part of the Copernicus Emergency Management Service. GloFAS provides extended-range (30 days) and seasonal probabilistic flood forecast for the major river basins in the world including the Brahmaputra basin in Bangladesh. GloFAS forecasts

could be used to complement the information available from the local observation network and forecasting models with probabilistic predictions at longer lead times. In this perspective, further work is necessary to evaluate the forecast skill of GloFAS in the Brahmaputra basin and identify its possible links with the different flood characteristics discussed in this paper.

## 7 Conclusions

The Brahmaputra is a massive river which is expected to response slowly due to monsoon rainfall in the downstream section of the basin located in Bangladesh. During the 2017 monsoon, the river at Bahadurabad behaved differently, with localised rainfall over the lower sub-basins of three tributaries (Teesta, Dharla and

Dudkumar rivers) contributing to river flow at the same time, leading to a sharp rise in water level. While the flood water level exceeded its previous record, discharge was not unprecedented, though both time series showed an unprecedented high frequency component. This suggests that the extreme (but not unprecedented) rainfall fell over a hydrological sweet spot of 3 adjacent tributaries: the Teesta, the Dharla and the Dudkumar. It was, therefore, the spatial distribution of rainfall which was important in driving the 2017 flooding. Arguably, long duration floods such

as the 1998 event, caused by long and widespread rainfall anomalies are more impactful in Bangladesh. However, floods such as those in 2017, with a rapid-increase in water level, are more difficult to forecast. This type of severe flood may occur in future, so flood forecasts need to skilfully represent the spatial distribution of rainfall events, their intensity and effects on river flows during the monsoon period. In addition, climate projections and attribution need to consider how both drivers will change in the future to adequately assess future flood hazard in the basin.



## Authors contribution

EMS and HLC assisted with concept development. The study was led, conceived and carried out by SH who analysed the data. EMS and HLC also assisted with analysis, interpretation along with AF, AGT. All the authors contributed to the writing, reviewing & editing.

**Competing interests:** The authors declare that they have no conflict of interest

## Acknowledgements

This study has been carried out as part of a PhD research supported by the Natural Environment Research Council and Department for International Development under the Forecasts for AnTicipatory HUManitarian action (FATHUM) project (grant number NE/P000525/1) and SHEAR Studentship Cohort programme (grant number NE/R007799/1). AGT acknowledges support from the National Centre for Atmospheric Science. The authors are thankful to the Bangladesh Meteorological Department and Bangladesh Water Development Board (BWDB) for providing observed meteorological and hydrological data. We are also grateful to European Centre for Medium-Range Weather Forecasts (ECMWF) for supporting the authors as visiting scientists to carry out the research work.

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

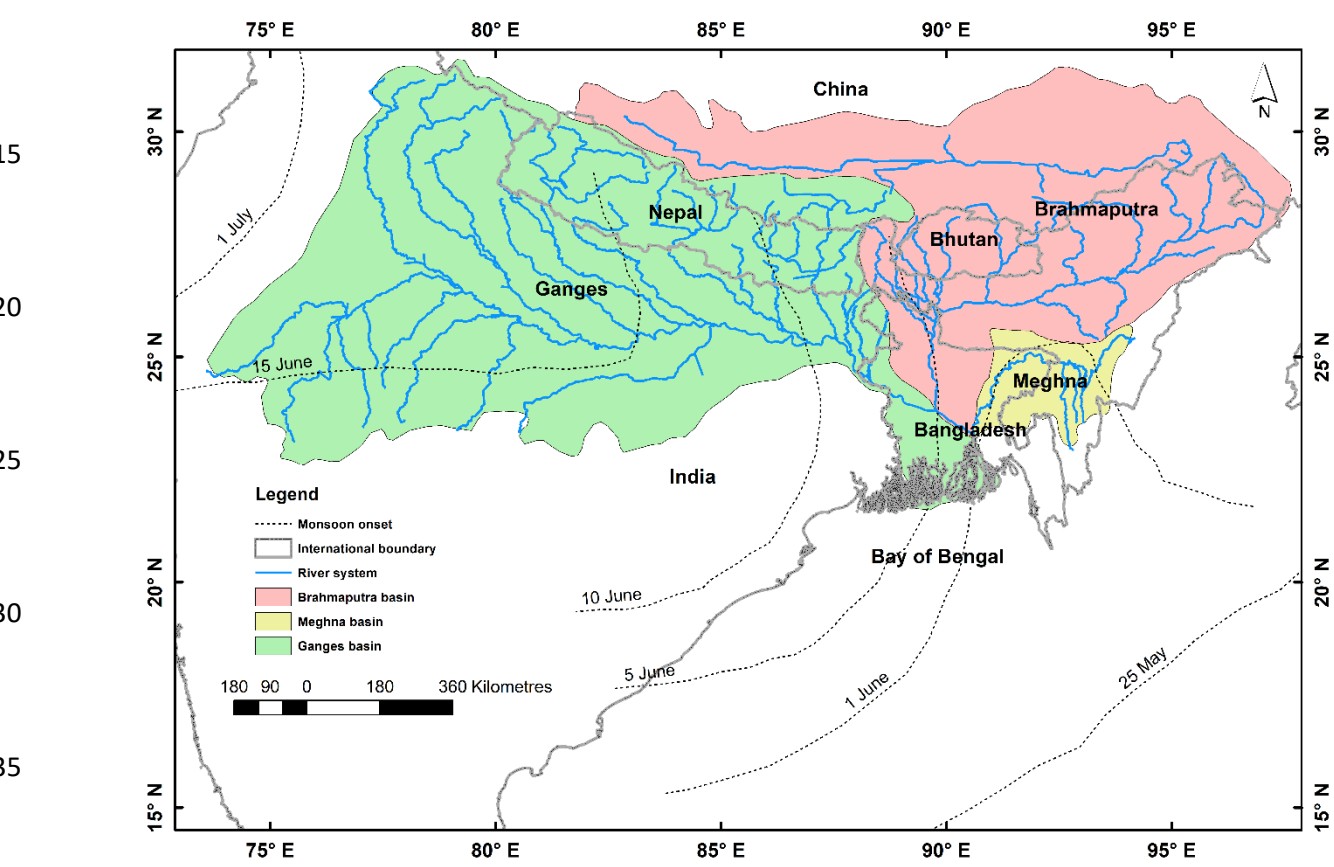

Figure 1: Monsoon onset normal dates towards the Ganges-Brahmaputra and Meghna basin (IMD, 2017a)

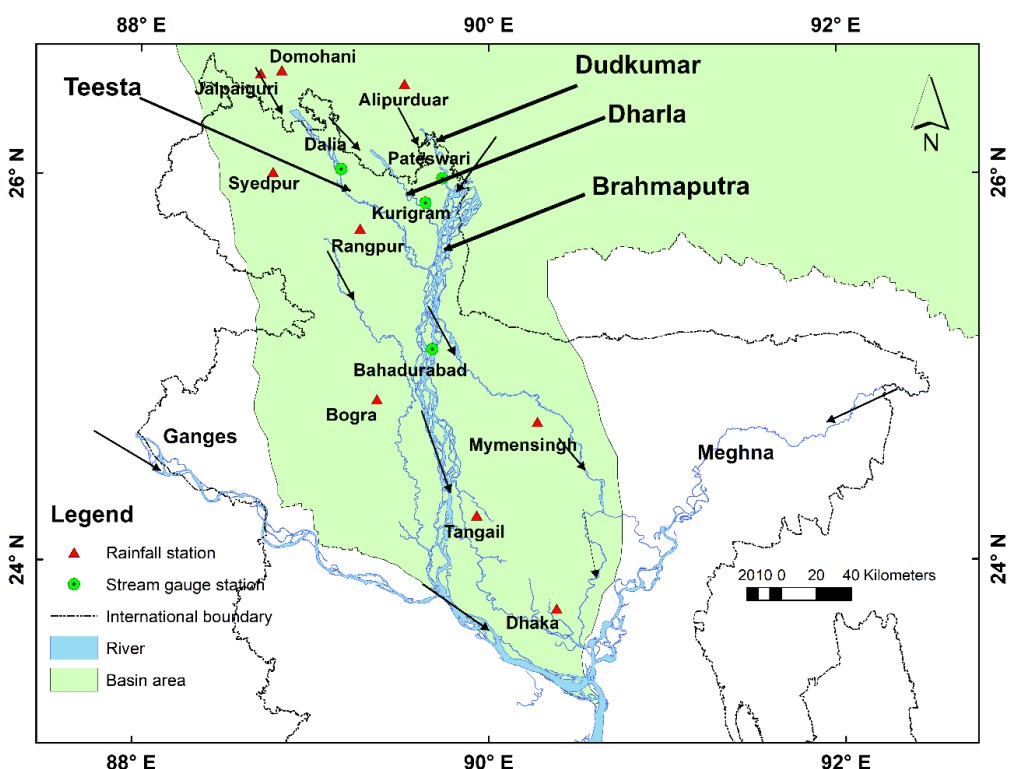

Figure 2: Basin area in Bangladesh with major river systems. The locations of the Bahadurabad stream gauge and other stream gauge and rain gauge stations are also shown (station locations data from FFWC).

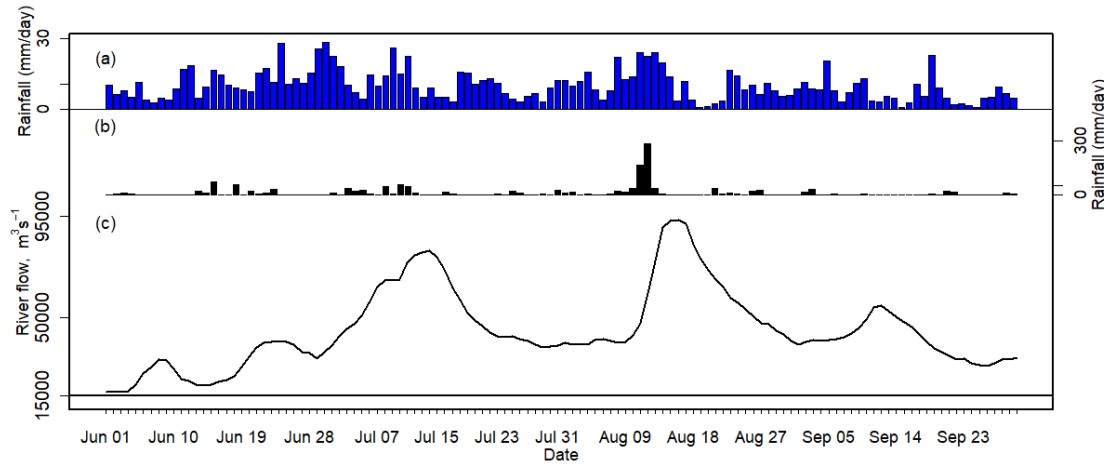

Figure 3: (a) basin-average rainfall (mm/day); (b) daily rainfall recorded by a rain gauge at Syedpur (mm/day) and (c) river flow of the Brahmaputra river at Bahadurabad during June to September 2017

15   (m$^3$/s).

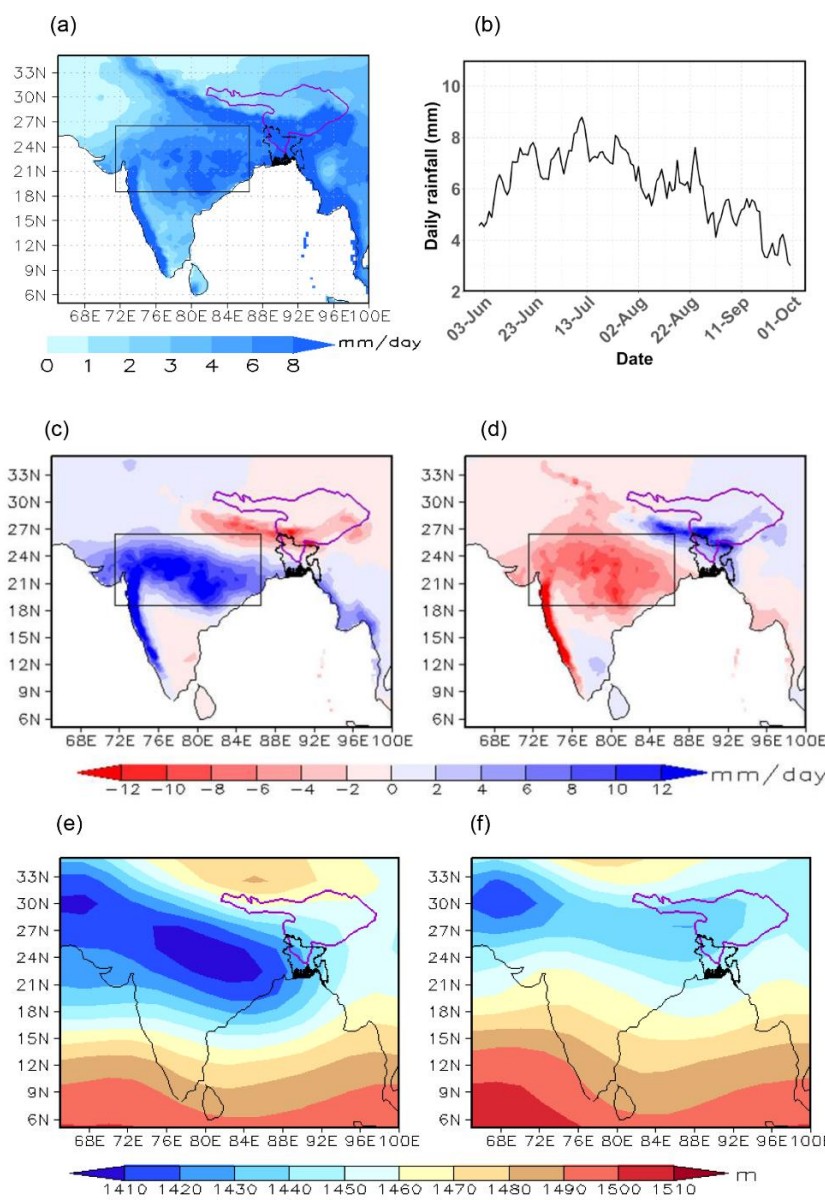

Figure 4: (a) distribution of mean daily JJAS rainfall and (b) basin-average JJAS rainfall of the Brahmaputra river basin over 1951-2007. Composite rainfall anomaly during (c) active and (d) break phases using standardized rainfall anomaly over the Indian monsoon core zone. Rainfall data are derived from JJAS Aphrodite 0.25° x 0.25° gridded data over the period 1951-2007. Composite 850 hPa geopotential height during (e) active and (f) break events based on the NCEP/NCAR reanalysis data. The purple line indicates the basin boundary of the Brahmaputra river.




Figure 5: The synoptic situation during two rainfall events in July and August 2017: 850 hPa geopotential height (m) anomaly and mean of absolute wind vector (a) 1-7 July, (b) 8-13 August; mean sea-level pressure anomaly (hPa) for (c) 1-7 July, (d) 8-13 August; anomaly in precipitable water (mm) for (e) 1-7 July and (f) 8-13 August (anomalies are estimated with respect to the base period 1987-2016



(a)

(b)

(c)

Figure 6: (a) The mean JJAS rainfall (mm/day) difference between La Niña and El Niño years over 1998-2017 years (based on TRMM 3B42 daily data); (b) Monthly ONI index of 3-month running mean SST anomalies in the Niño-3.4 region (5°N-5°S, 120°-170°W), where blue and red lines indicate the -0.5°C and +0.5°C thresholds, respectively (Source: NOAA, 2019); (c) BSISO mode shown by green line for PC1 and black line for PC2 for 2017. Blue and red lines indicate +1 and -1 thresholds respectively. The grey shading in the background indicates a defined BSISO event (amplitude exceeding the threshold) (Source: IPRC/SOEST, 2019)

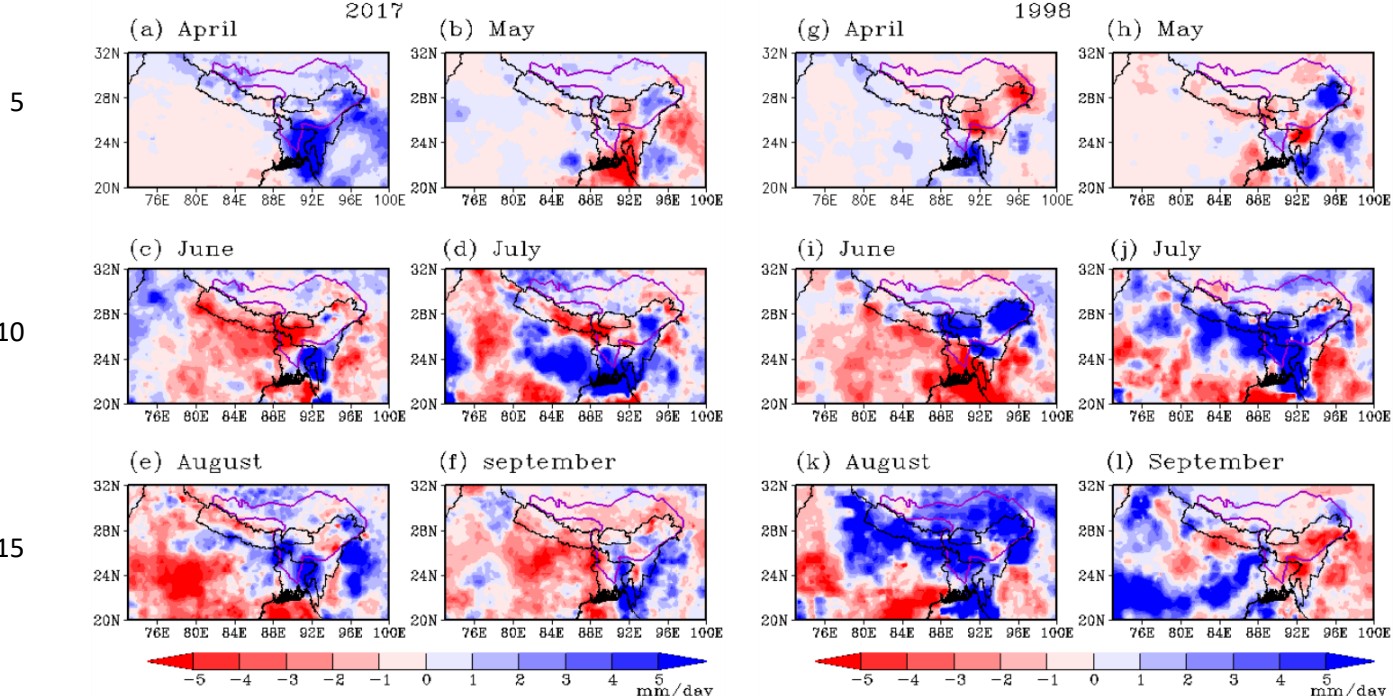

Figure 7: Distribution of mean monthly rainfall anomaly (mm/day) for each month from April to September, left panel 2017 and right panel 1998 (anomaly is estimated using a climatology period of 1999 to 2016 and based on TRMM 3B42 daily).



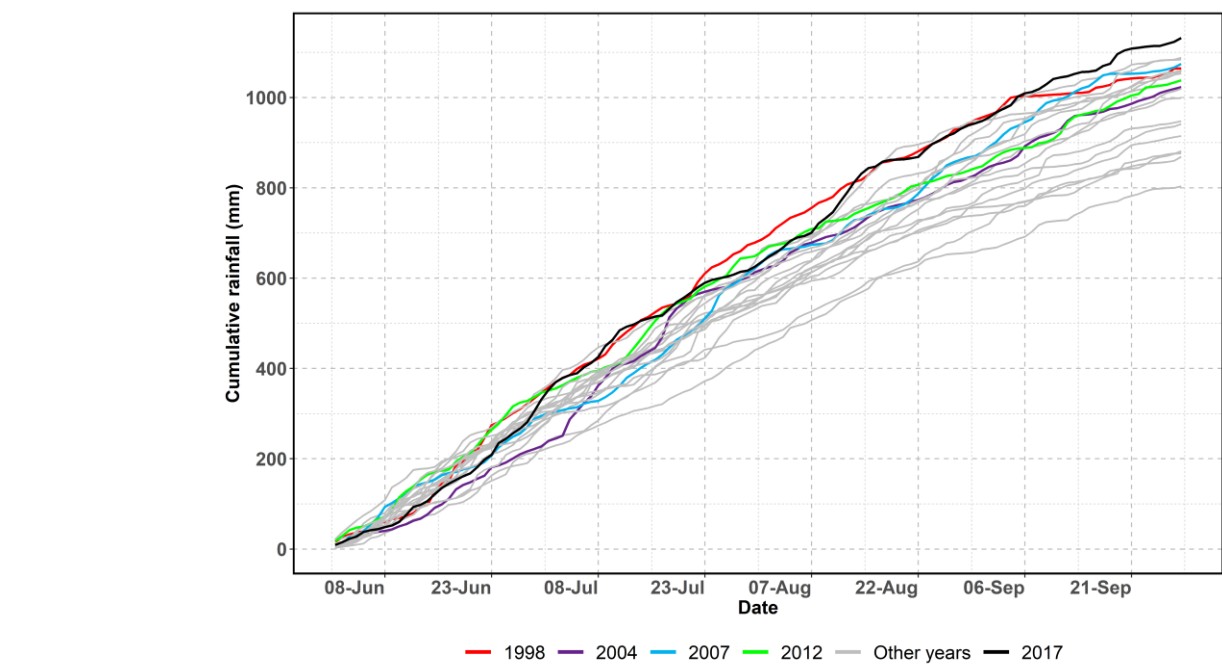

Figure 8: Cumulative rainfall (June to September) of the Brahmaputra basin with flooding years highlighted (1998, 2004, 2007 and 2012 are severe flooding years in the Brahmaputra basin in Bangladesh). The 2017 flood is shown by the black line.

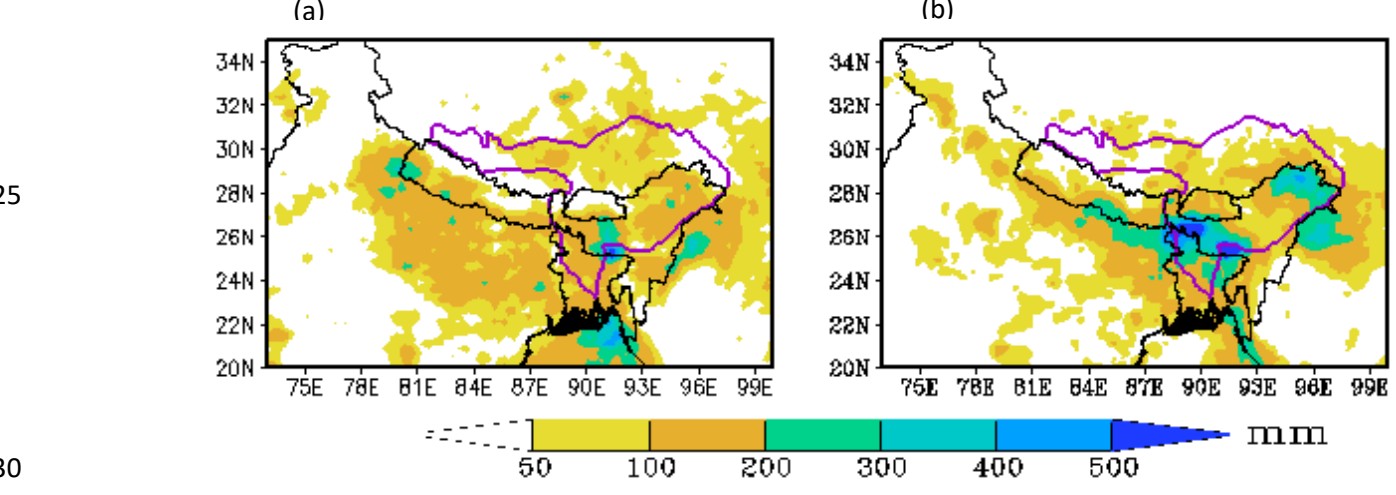

Figure 9: Accumulated rainfall (mm) (a) 1-7 July 2017 and (b) 8-13 August 2017 (based on TRMM 3B42 daily data)





10    `

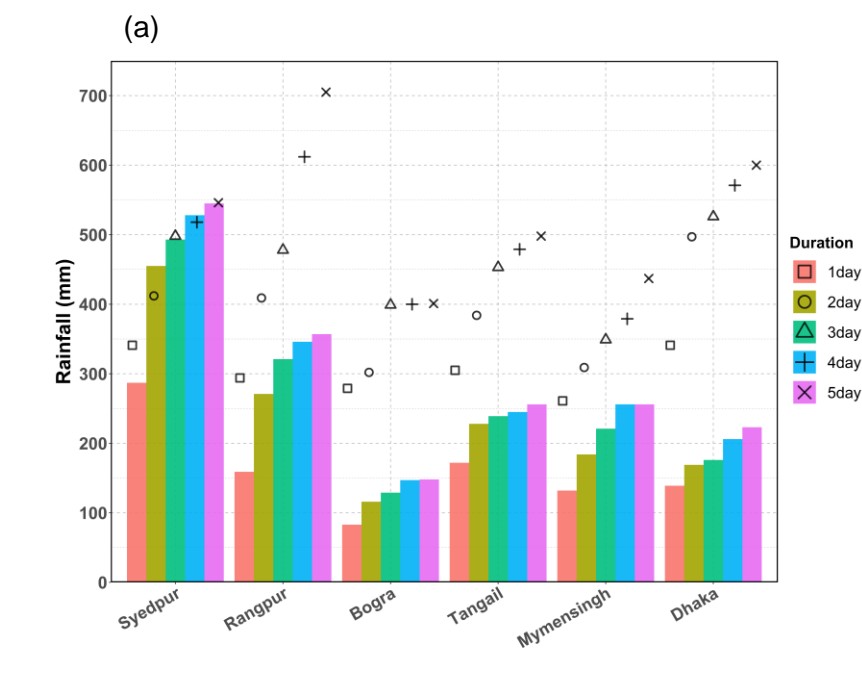

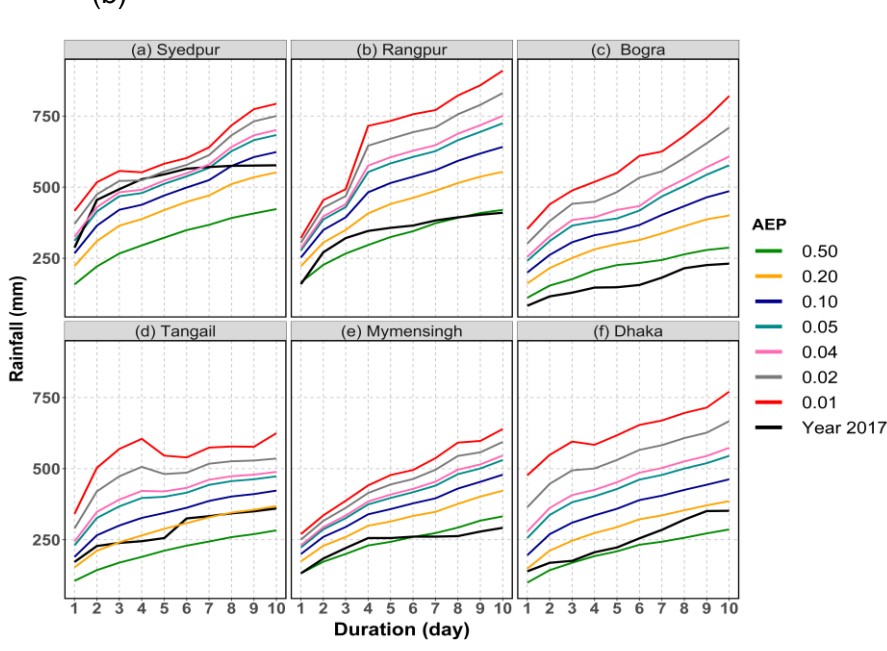

Figure 10: (a) Observed maximum rainfall (mm) of 1-5 day duration between 1987-2016 and 2017 and (b) Depth-duration-frequency curves of 6 rain gauges in the Brahmaputra basin in Bangladesh coloured by the Annual Exceedance probability (AEP).



Figure 11: (a) Full hydrograph prepared by observed daily water level (m) at Bahadurabad gauge station 1987 to 2017; (b) The low frequency component (A6) is a slowly changing element obtained from wavelet transform of a six-level decomposition of daily water level. (c) High frequency component (D4) of wavelet transform at 16-days variation (other high frequency components such as D1, D2, D3 are not shown here). Similarly, Figure 11d, 11e and 11f show river flow (m³/s) hydrograph, low frequency component and high frequency component of wavelet transform respectively.

Figure 12: Scatter diagrams of 3-day mean water level rise (cm/day) versus water level (m) during the monsoon period in different years for the 1987-2007 period at: (a) Bahadurabad (Brahmaputra), (b) Kurigram (Dharla), (c) Pateswari (Dudkumar) and (d) Dalia (Teesta). The black vertical line indicates the water danger level while horizontal lines show 80th, 90th and 95th percentiles of rise in water level.







Figure 13: Flood duration in days above danger level from 1987 to 2017 at: (a) Bahadurabad (Brahmaputra), (b) Kurigram (Dharla), (c) Pateswari (Dudkumar) and (d) Dalia (Teesta)

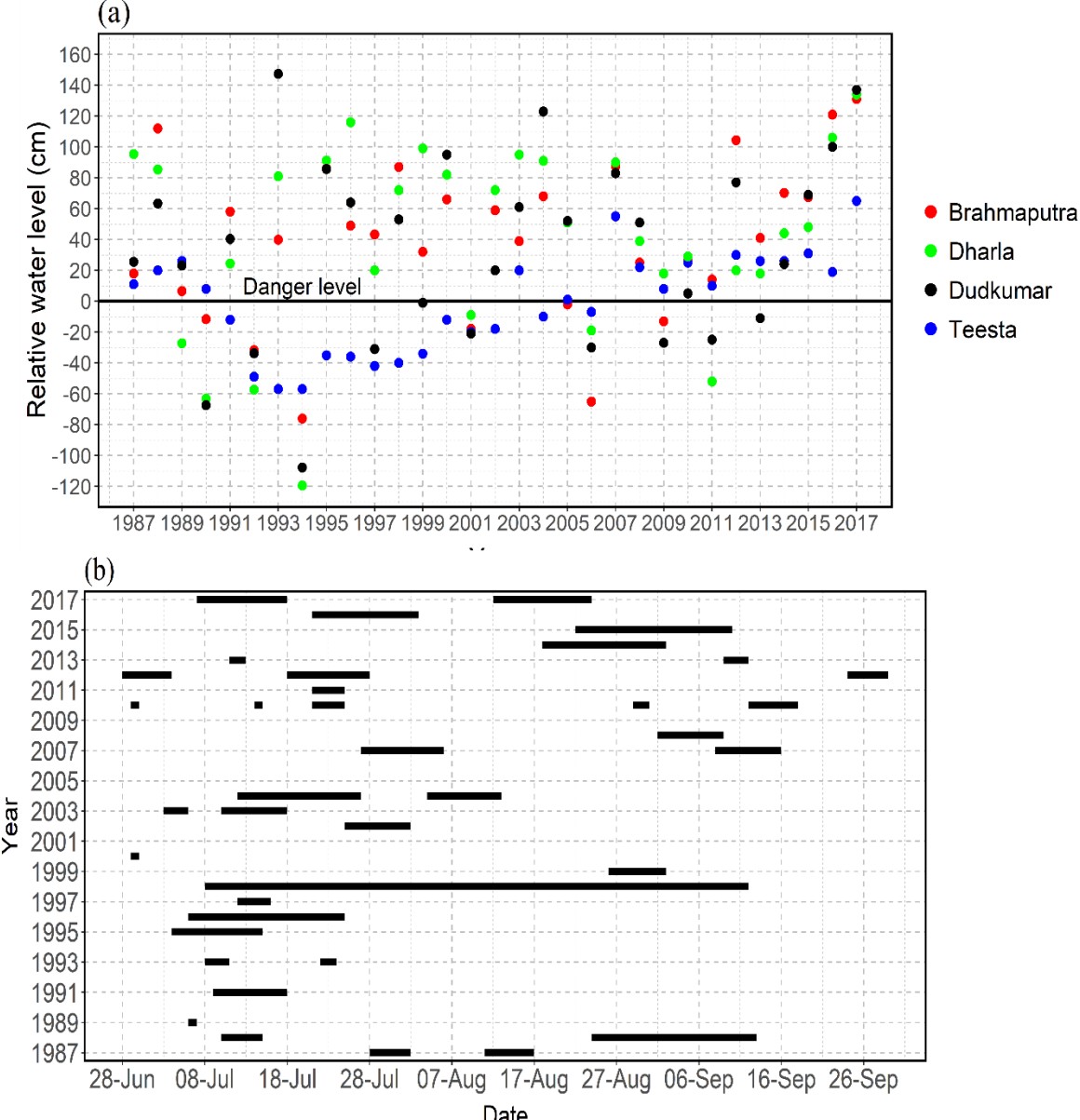

Figure 14: (a) Relative water level above/below danger level (cm) for the Brahmaputra and three tributaries; (b)
Dates indicated show when the water level exceeds the danger level at the Bahadurabad station of the
5    Brahmaputra river





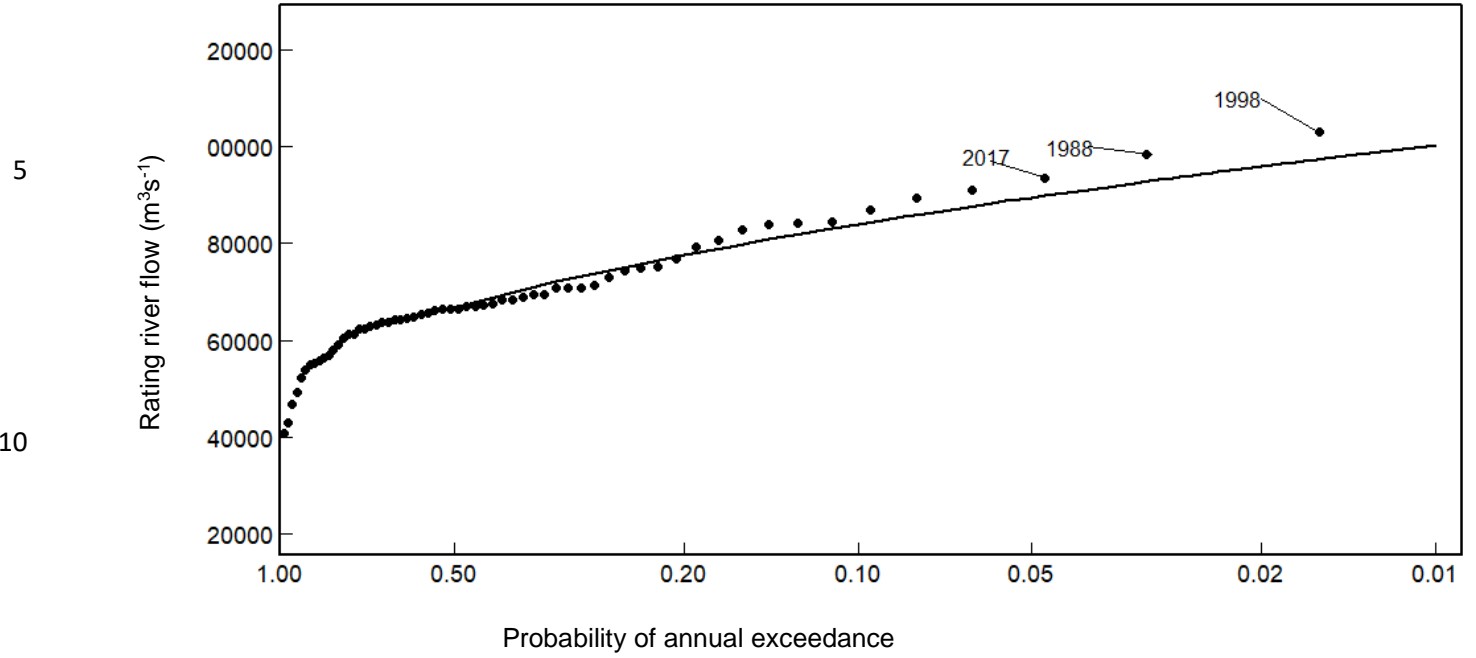

Figure 15: Exceedance probability of annual maximum flow (m$^3$/s) estimated from rating curve of the Brahmaputra river at Bahadurabad gauging station