# Peer review of "Hydrometeorological drivers of the 2017 flood in the Brahmaputra basin in Bangladesh"

_Hydrology and Earth System Sciences, 2019_

## Referee Comment (RC1) · Anonymous Referee #1 · 28 Jul 2019

This manuscript takes a detailed look at the flood event in 2017 in Bangladesh. The characteristics of the flood event is analysed, the hydrometeorological drivers are investigated. Flood events in the Indian Monsoon areas always cause a large amount of economic loss every year, and even cause numbers of death in some severe cases. The focus of this research is meaningful for this dense populated region, the results revealed could provide useful information for future understanding of this natural hazard. However, still the manuscript needs to be much improved in a series of aspects before publishing in HESS. I recommend a major revision.

Major:
1. According to the Abstract the primary conclusion of this manuscript is "The study concludes that the location and magnitude of extreme rainfall are key drivers controlling on the characteristics of the Brahmaputra floods." These sentences emphasis the location and magnitude of extreme rainfall are critical findings (have to say that the selling point is plain). However, according to the Introduction part, the major purpose of this manuscript is to find out the major hydrometeorological drivers behind the flood in 2017. Hydrometeorological drivers includes much more information than the location and magnitude of extreme rainfall only. Either specify introduction part about which drivers you are particularly look at and why, or rewrite the abstract in a more concise way.
Clarifying the major purposes are also important for the manuscript structure. So far, the structure of this manuscript is a bit loose, including too many figures and some abundant/not-well-organized information. For example, the manuscript can majorly focus on three aspects: the flood event, the flood-triggering rainfall, the rainfall-triggering synoptic systems, and arranged them in a way that each finding is highly connected, it will be a good manuscript.

2. Abstract should include the very firm findings with solid information and statement mentioned in the manuscript. For example, in the Abstract, "This heavy rainfall was associated with a northward shift of the monsoon trough, creating active monsoon conditions in the Brahmaputra basin." This sentence is obscure. As mentioned in the manuscript, the monsoon trough has basically two positions with active and break events positions according to the historical reanalysis data. However, the "northward shifting" is compared to active or break positions? It is a very critical information for the mechanisms for the flood event in 2017. Otherwise, the climatology of the monsoon position can be shown, and then you can say there is a northward movement of the monsoon position during the 2017 flood event. Similar problem is also shown in 5.1.2 last paragraph. Meanwhile, since it is the major finding of the manuscript and highlighted in Abstract, the related information in the major body can not say in this way: in 5.1.2 last paragraph "suggesting it may have played a role in the synoptic activity leading to the 2017 Brahmaputra flooding". It is a very weak statement, what kind of role and how it plays. Both of them should be illustrated clearly. Otherwise, remove this major finding from the abstract.

3. Flood 1998 is used for some comparison with 2017 flood, some background should be included in the Introduction/methodology part of this 1998 flood. Also, 1988 is mentioned several times, with similar characters with 2017 and 1998? It will be good to include some details on this flood 1988 too.

4. In the Introduction part, the research gap and the necessity of doing this research are not well illustrated. Since you are going to figure out at least three aspects of this flood event, the introduction should not only introduce the research gap in "synoptic system". And why the flood in 2017 is so outstanding from the historical flood events, what we can learn from investigating this flood event?

5. Shorten the words related to climate change and future forecasting. This research investigates the flood even in 2017 July and August in detail, with some analyses on the inter-annual variability. It is hard to have strong implication for the long-term (at least 30 years) climate change. Meanwhile, it is only a case study comparing with another historical flood case; it is hard to improve future forecasting that mostly uses statistical or modelling methods based on clear mechanisms, fine spatial and temporal resolution. Thus, all the implications for climate change and future forecast should be written carefully, and should be based on solid and related findings, otherwise, just remove them to make the manuscript short, concise and focused.

Minor:
1. "2. Characteristics of the Brahmaputra basin"
Keep the decimal places consistent in the latitude and longitude number.
2. remove the full name of GEV in line 32 at page 6 because it has been mentioned in the upper paragraph
3. Fig. 3b why choose the rain gauge at Syedpur as a representative rain gauge station (there are around 9 stations within the basin according to Fig. 2)?
4. Fig. 4e, f, are they also come from the 1951-2007 years?
5. Figure 11a, the DL means danger level should be included in the caption.
6. 3.2.2 The time period of used TRMM data should be included here; 3.2.3 The time and spatial resolution of NCEP reanalysis data should be included here.
7. Figure 4 included the purple line indicating the basin boundary of the Brahmaputra river; The purple line should be shown in Figure 1 as well to highlight the river basin you are going to look at. Figure 2 can be merged into Figure 1 as 1b with a box zooming in/out to show the location relationships with Figure 1. Also, the foothills of the Himalaya and the Assam region are mentioned many times in the manuscript and should be marked in these figures.
8. Figure 5a, b: the legend of wind vectors should be included.

---

## Referee Comment (RC2) · Anonymous Referee #2 · 4 Aug 2019

**Review of article entitled "Hydrometeorological drivers of the 2017 flood in the Brahmaputra basin in Bangladesh" for potential publication in HESS.**

This paper conducted detailed analyses on the 2017 floods in Brahmaputra river basin, with the hope to answer questions regarding hydrometeorological drivers of such floods different from historical ones (rapid rise of water). I found the paper generally well written; the materials are interesting and overall comprehensive. I think this paper can be publishable in HESS eventually, but before it can move forward, I am providing several major and minor comments for authors to improve their manuscript. My major complaint is that the study probably focused too much on the meteorological side while the hydrological (land surface storage) side of drivers was largely ignored. For example, antecedent soil moisture could be very important in explaining the unique flood response in 2017, but it was bypassed in this paper (see my detailed comment below regarding Fig. 7). In addition, I agree with the other reviewer that this paper contains many repetitive information, which makes the presentation of this paper loose and can distract the readers' attentions. I listed several concrete examples below, but note that not all of them have been listed. So, there is the need for the authors to carefully and thoroughly revise their English writing and story-telling flows, before this paper can move forward.

Major:
- I feel this paper spent a lot of efforts on "meteorological" drivers (which is good and comprehensive). However, to complete its "hydrometeorological drivers" claim in the title, some soil moisture analyses/discussions may be necessary. For example, looking at Fig. 7: it seems to me that the rapid rise of water level for the 2017 floods (different from 1998) may be due to the high antecedent soil moisture in 2017, which is resulted from the very wet spring (April) in 2017. This may have led to the rapid rise of water in 2017. By contrast, in 1998, July and August rainfall anomalies were higher but the prior spring seems to be dry (especially in the lower Brahmaputra), which leaves large room for soil storage and thus leading to much slower flood water rise. I suggest the authors to present some soil moisture analyses, which can help complete their story.
- Authors are suggested to reference a very similar analysis conducted in the US (https://journals.ametsoc.org/doi/full/10.1175/JHM-D-18-0038.1), where the researchers found the unique response of a flood was due to similar reasons as found in your study. But they also have provided more detailed analyses on antecedent soil moisture as well as flood celerity in the tributaries above the flooded location (which is also similar to this study). Authors are suggested to gather some soil moisture data (from remote sensing) to perform some analyses to improve their story-telling. Or at the very least, these important "hydro" drivers need to be carefully discussed by the paper.

Minor:
- Fig. 2: I found several arrows do not have any associated texts and were placed in wrong place maybe. Please revise the figure to make sure arrows are correctly drawn;
- Fig. 3: why choosing the rain gauge at Syedpur but not others?
- Fig. 4 caption: change to "over the Indian monsoon core zone (rectangular box)";

- Fig. 5 caption: missing a bracket;
- Fig. 8: the starting point of the calculation starts from June. Is it possible to start from spring (e.g. April)? The reason is because the spring rainfall anomaly is important for understanding the antecedent basin wetness condition before flooding (recall my Major Comment earlier).
- P1L19: change "but" to "and"; no transition needed here;
- P1L22: the sentence "Water level and river flow time series …" should be better placed in L28 before "The wavelet analysis";
- P3L2: change "the river flow" to "the water level"
- P3L5: change "consider" to "analyze"
- P6L9: change to "study of Zhang et al. (2017)"
- English presentation of this paper contains several repetitive phrases. For example, (1) P6L2 and P6L3; can use "it" to replace "the Daubechies wavelet function"; (2) P6L13 and P6L24: suggest authors to ask help from native speakers to improve English presentation; here only limited examples are provided but more places need to be thoroughly revised;
- P6L32: no need to spell out "GEV" again;
- P8L1: change "shifting" to "shift"
- The English presentation of this paper can be much better simplified. For example, P8L23 can be changed to "An El Nino (La Nina) state is defined when ONI exceeds 0.5 degree C (below -0.5 degree C)". Same apply to P7L14;
- P8L28: many of these abbreviations have been defined before, no need to define again. Please check throughout the paper, there are many such cases;
- P9L9: remove "anomalies"! It appeared again later and this use is incorrect;
- P16L9: MJO again; no need to spell out

---

## Author Comment (AC1) · 3 Oct 2019

**Response to the comments of Reviewer 1**

We are very much thankful to anonymous Reviewer 1 for his time to review the article and provide valuable comments. These comments are useful to improve the overall quality of the article. We will address these in the revised manuscript and accordingly our responses to each comment is given below. We have separated some of the major comments from Reviewer 1 into two separate parts to respond more clearly. We marked our replies in a blue font, while original reviewer comments are presented in a black font.

**Major:**

*RC1 (a): According to the Abstract the primary conclusion of this manuscript is "The study concludes that the location and magnitude of extreme rainfall are key drivers controlling on the characteristics of the Brahmaputra floods." These sentences emphasis the location and magnitude of extreme rainfall are critical findings (have to say that the selling point is plain). However, according to the Introduction part, the major purpose of this manuscript is to find out the major hydrometeorological drivers behind the flood in 2017. Hydrometeorological drivers includes much more information than the location and magnitude of extreme rainfall only. Either specify introduction part about which drivers you are particularly look at and why or rewrite the abstract in a more concise way.*

**Response:**

AC1 (a): We agree that there are many hydrometeorological drivers that characterize a flooding event. In the current study, we focus on the rainfall situations that trigger flooding as well as monthly rainfall anomaly to find variation from the long-term average. We also analysed two important hydrological elements- water level and discharge to study hydrological response. We will rewrite the abstract to be clear in specifying which drivers are considering and why. We will also include soil moisture in this current study.

*RC1 (b): Clarifying the major purposes are also important for the manuscript structure. So far, the structure of this manuscript is a bit loose, including too many figures and some abundant/not-well-organized information. For example, the manuscript can majorly focus on three aspects: the flood event, the flood-triggering rainfall, the rainfall-triggering synoptic systems, and arranged them in a way that each finding is highly connected, it will be a good manuscript.*

**Response:**

AC1 (b) We agree with the suggestion of the reviewer of making the structure of the manuscript more effective for our purpose. Our main purpose is to study the important meteorological and hydrological drivers which were primarily responsible for the severity of the 2017 monsoon flooding in the Brahmaputra basin compared with previous years. This is why we have analysed in detail different hydrometeorological aspects of the 2017 flood event and compared those with previous event as 1998. We think that our manuscript needs to include also an analysis of the large scale drivers and subseasonal conditions during the monsoon in addition to the main aspects mentioned by the reviewer above (the flood event, the flood-triggering rainfall and the rainfall-triggering synoptic systems). We also think that the structure of the analysis follows a logical path, as for example in Section 5.1, starting with the more local meteorological characteristics (synoptic situation) and moving to larger-scale drivers at the end (e.g. ENSO, BSISO/MJO indices). We studied the monthly rainfall characteristics and finally the flood-generating rainfall events of 2017, before moving to the hydrological analysis of the 2017 floods. We believe that in order to diagnose fully the hydromet drivers we need to elucidate the evidence and include all these aspects, and we would dispute that these are too many figures. However, we merged Figure 1 and 2 together to lower the number of figures with respect to the original manuscript, but we have not found any other superfluous figure. Moreover, we are revising the results section (section 5) to summarize more the information considering the suggestion of the reviewer to focus on some main aspects in the most concise way as possible.

*RC2 (a): Abstract should include the very firm findings with solid information and statement mentioned in the manuscript. For example, in the Abstract, "This heavy rainfall was associated with a northward shift of the monsoon trough, creating active monsoon conditions in the Brahmaputra basin." This sentence is obscure. As mentioned in the manuscript, the monsoon trough has basically two positions with active and break events positions according to the historical reanalysis data. However, the "northward shifting" is compared to active or break positions? It is a very critical information for the mechanisms for the flood event in 2017. Otherwise, the climatology of the monsoon position can be shown, and then you can say there is a northward movement of the monsoon position during the 2017 flood event.*

**Response:**

AC2 (a): We agree that the sentence mentioned by the reviewer from our abstract was a bit confusing ("*This heavy rainfall was associated with a northward shift of the monsoon trough, creating active monsoon conditions in the Brahmaputra basin*") and we are suggesting a clearer version of it. In fact, in that sentence we were referring in particular to the consequences of this northward shift of the monsoon trough in the Brahmaputra basin; so, by the wording "*active monsoon conditions in the Brahmaputra basin*" we were actually referring to a break condition in general meteorological terms (i.e. over the monsoon core zone, central India) bringing more rainfall to the Brahmaputra basin. However, we did not notice that there was a possible misunderstanding, given that we were using the wording "active monsoon condition" which can be misleading. So, we have rewritten the sentence as:
"*These heavy rainfall events in the Brahmaputra basin were associated with a northward shift of the monsoon trough.*"

In the original manuscript (Section 5), we had provided the position of the monsoon trough based on the climatology as well as for the 2017 floods. The climatology of the monsoon trough is shown in Figure 4e and 4f and Figure 5a and 5b represent trough position during the 2017 flood periods.

RC2 (b): *Similar problem is also shown in 5.1.2 last paragraph. Meanwhile, since it is the major finding of the manuscript and highlighted in Abstract, the related information in the major body cannot say in this way: in 5.1.2 last paragraph "suggesting it may have played a role in the synoptic activity leading to the 2017 Brahmaputra flooding". It is a very weak statement, what kind of role and how it plays. Both should be illustrated clearly. Otherwise, remove this major finding from the abstract.*

AC2 (b): We understand the concern of the reviewer, and we have provided more elements now to support that statement in Section 5.1.2 and illustrate the role of the Boreal Summer Intraseasonal Oscillation (BSISO) more clearly. The northward propagation of the BSISO is linked with the short-term variability of the monsoon in off-equatorial trough regions (further away from the equator). On the other hand, the propagation of the Madden-Julian Oscillation (MJO) is eastward along the equator (Kikuchi et al., 2012; Lee et al., 2013). The strength of MJO and BSISO is described by the first two principle components of the PC analysis (see Figure S4, phase diagram of MJO and 6 (c) of the original manuscript). We are adding a panel to previous Figure 6, as reported in the following Figure AC1 showing the strength of MJO and BSISO during the 2017 monsoon (where we have added a panel, with the principal components of MJO). Therefore, the evolution of BSISO (northward propagation) in 2017 suggests that this has an influence on the northward propagation of the monsoon. As shown in Figure AC1(b), there is a strong BSISO event from end of July to mid-August 2017 (see grey shaded area, i.e. PC components over the threshold) which is likely to create favourable synoptic activity for the Brahmaputra flooding of 2017, given the temporal correspondence of this event with the northward propagation of the monsoon trough. So, we have modified the sentence mentioned by the reviewer accordingly. The new sentence is reported below:

"The BSISO mode shows strong northward propagation between end of July and mid-August during the 2017 monsoon (Figure 6c, grey area). This is likely to contribute to favourable synoptic activity for the Brahmaputra flooding of 2017, given the temporal correspondence of this event with the northward propagation of the monsoon trough." However, the influence of the BSISO intraseasonal oscillation mode on the monsoon rainfall and flooding in the Brahmaputra basin may need further investigation as mentioned in our recommendation to understand more the quantitative links.

[Figure]

Figure AC1: Principal components (a) MJO and (b) BISOS in 2017

*RC3: Flood 1998 is used for some comparison with 2017 flood, some background should be included in the Introduction/methodology part of this 1998 flood. Also, 1988 is mentioned several times, with similar characters with 2017 and 1998? It will be good to include some details on this flood 1988 too.*

**Response:**

AC3: We will make this flood history context clearer in the revised manuscript and provide further explanation on the 1998 flood as the reviewer requests.

We compared the flood characteristics such as flooding duration, water level rise, peak water level and discharge considering the 2017 floods and other major events within the previous 30 years. The flood in 1998 was particularly important to mention as it was the longest duration flood with highest impact in the monsoon flooding history of Bangladesh. The country remained flooded for two and half month and 68% of the country was affected by the floods (FFWC, 1998). The Brahmaputra experienced flooding along with the other two river basins, Ganges and Meghna, and the flood peaks of the three rivers were synchronised. The water levels in the Brahmaputra river at Bahadurabad gradually crossed the danger level on 8 July in 1998 and continued to flow over this level for the next 66 days. Therefore, it is important to focus on the 1998 flooding event and compare it with the recent flooding of 2017 to understand how two different hydro-meteorological conditions may cause flooding in the Brahmaputra river basin in Bangladesh. Not only the 1998 was a huge flood, it is also essential to understand the different natures of the floods in the Brahmaputra basin in order to ensure effective preparedness based on hydromet understanding. Otherwise, there is an assumption that future floods will be like 1998.

*RC4. In the Introduction part, the research gap and the necessity of doing this research are not well-illustrated. Since you are going to figure out at least three aspects of this flood event, the introduction should not only introduce the research gap in "synoptic system". And why the flood in 2017 is so outstanding from the historical flood events, what we can learn from investigating this flood event?*

**Response:**

AC4: We agree with the reviewer comment on the fact that "the introduction should not only introduce the research gap in "synoptic system". So, we will provide a clearer statement of our research niche in the introduction, specifying our research questions, as for example: (i) Why the flood in 2017 is so outstanding from the historical flood events? (ii) what we can learn from investigating this flood event?.

There are several case studies available on some past floods in Bangladesh such as 1988, 1998, 2004 and 2007. Most of the studies used monthly rainfall or seasonal rainfall to describe the flooding characteristics of the particular year and focused less on the other meteorological aspects such as synoptic situation or event-based rainfall that are linked with the flooding. There is a lack of studies on the rainfall events that trigger flooding in the Brahmaputra basin in Bangladesh that we address in our study.

Secondly, available literature focused only on a few characteristics of the floods, such as flood duration and peak water level or river flow for a particular flooding year. No previous study has investigated the river water level rise. Similarly, soil moisture was not considered in any study to better understand flood drivers, and we have added some new elements to address this gap. We used wavelet analysis to study hydrological time series for identification of short-term variation and annual cycle which has not been applied before to study the flood behaviour of the Brahmaputra river. This helped to identify the variation of the dominant component which influence on the floods compared to previous years floods.

*RC5: Shorten the words related to climate change and future forecasting. This research investigates the flood even in 2017 July and August in detail, with some analyses on the inter-annual variability. It is hard to have strong implication for the long-term (at least 30 years) climate change. Meanwhile, it is only a case study comparing with another historical flood case; it is hard to improve future forecasting that mostly uses statistical or modelling methods based on clear mechanisms, fine spatial and temporal resolution. Thus, all the implications for climate change and future forecast should be written carefully, and should be based on solid and related findings, otherwise, just remove them to make the manuscript short, concise and focused.*

**Response:**

AC5: We agree with the reviewer that the implications for climate change and future forecasting should be written carefully. So, we will shorten the words related to climate change and future forecasting as asked by the reviewer. Also, we will reword the paragraph on these aspects to demonstrate more clearly how the findings and evidence provided in our manuscript can still contribute to these aspects, in terms of recommendations for further research.

Climate change projection shows that future potential changes in the monsoon precipitation pattern. Some studies demonstrate the attribution of the impact of climate change on the flooding of the Brahmaputra basin (Philip et al., 2019; Uhe et al., 2019). Flooding in the Brahmaputra basin is an annual phenomenon. Therefore, it may be difficult to attribute the particular flooding event to the climate change. Here, we have mentioned some implications of climate change from attribution perspective. Climate change projection mostly consider long-term changes such as changes in mean seasonal

precipitation. According to our study, two different patterns of rainfall may cause flood severity in the basin. Therefore, the sub-seasonal scale rainfall events which trigger the floods are essential to consider in the climate change studies.

Finally, we would like to respond to the last major reviewer comment "*it is only a case study comparing with another historical flood case; it is hard to improve future forecasting that mostly uses statistical or modelling methods based on clear mechanisms, fine spatial and temporal resolution.*" In this regard we argue that if we do not understand the base line drivers then we cannot assess their representativeness in future modelling. This is also necessary to improve the understanding of the drivers that can guide flood forecasters to better interpret the quantitative forecasts about an upcoming flood event and assess the situation in real-time according to prevailing weather conditions. A modelling approach is also used to forecast floods in Bangladesh, but given the well-known model and data uncertainties forecasters use to complement model outputs with expert-based analysis of the hydro-meteorological situation. Also, we would like to stress the importance of this complementary information in the case of transboundary river basins, as the Brahmaputra, where limited hydro-meteorological data are available due to institutional barriers.

Minor:

RC 1. "2. Characteristics of the Brahmaputra basin"
Keep the decimal places consistent in the latitude and longitude number.
AC1: we change into consistent in the latitude and longitude number. (83.00°E-96.50°E and 24.50°N-30.25°N).
RC2. remove the full name of GEV in line 32 at page 6 because it has been mentioned in the upper paragraph

AC2: We remove full name of GEV

RC3: Fig. 3b why choose the rain gauge at Syedpur as a representative rain gauge station (there are around 9 stations within the basin according to Fig. 2)?

AC3: The rainfall distribution shows heavy rainfall occurred lower sub-basins near international border. Syedpur is located towards the north of the country where heavy rainfall occurred during the 2017 flood event. Therefore, it has been presented as a representative gauge to capture the rainfall event.

RC4. Fig. 4e, f, are they also come from the 1951-2007 years?
AC4: For the Fig. 4e, f same time span (1951-2007) has been used.

RC5. Figure 11a, the DL means danger level should be included in the caption.

AC5: Danger level is added in the caption

RC6: 3.2.2 The time period of used TRMM data should be included here; 3.2.3 The time and spatial resolution of NCEP reanalysis data should be included here.

AC6: TRMM data from 1998 to 2017 has been included. Similarly, spatial resolution of NCEP reanalysis has been included in the text- "The spatial resolution of NCEP reanalysis NCEP/NCAR Reanalysis 1 is 2.5 degree with the coverage 90N - 90S".

RC7. Figure 4 included the purple line indicating the basin boundary of the Brahmaputra river;

The purple line should be shown in Figure 1 as well to highlight the river basin you are going to look at. Figure 2 can be merged into Figure 1 as 1b with a box zooming in/out to show the location relationships with Figure 1. Also, the foothills of the Himalaya and the Assam region are mentioned many times in the manuscript and should be marked in these figures.

AC7: The Himalayan mountain system is classified into three sub-regions – (i) High altitude region of western Himalaya (ii) Eastern Himalaya and (iii) Himalayan foothills (Tiwari and Joshi, 1997). According to suggestions, we need to provide Himalayan foothills in the map. Therefore, we have revised Figure 1 accordingly. Now, we provide elevation based on the SRTM DEM to relate Himalayan foothills with the Himalaya mountain system (Figure RC 2). Figure 1 and 2 of the original article have been combined to make a single Figure as follows with the highlighting Himalayan foothills and Assam region.

[Figure]

Figure AC2: (a) Monsoon onset normal dates towards the Ganges-Brahmaputra and Meghna basin. The purple line indicates the basin boundary of the Brahmaputra river (b) Basin area in Bangladesh with major river systems. The locations of the Bahadurabad stream gauge and other stream gauge and rain gauge stations are also shown. The small arrows in the map show the direction of flow.

RC8. Figure 5a, b: the legend of wind vectors should be included.

AC8: Legend of wind vectors are included.

FFWC: Annual Flood Report, Flood Forecasting and Warning Centre, Bangladesh, 1998.

Kikuchi, K., Wang, B., and Kajikawa, Y.: Bimodal representation of the tropical intraseasonal oscillation, J Climate Dynamics, 38, 1989-2000, https://doi.org/10.1007/s00382-011-1159-1, 2012.

Lee, J.-Y., Wang, B., Wheeler, M. C., Fu, X., Waliser, D. E., and Kang, I.-S.: Real-time multivariate indices for the boreal summer intraseasonal oscillation over the Asian summer monsoon region, Climate Dynamics, 40, 493-509, https://doi.org/10.1007/s00382-012-1544-4, 2013.

Philip, S., Sparrow, S., Kew, S. F., van der Wiel, K., Wanders, N., Singh, R., Hassan, A., Mohammed, K., Javid, H., Haustein, K., Otto, F. E. L., Hirpa, F., Rimi, R. H., Islam, A. K. M. S., Wallom, D. C. H., and van Oldenborgh, G. J.: Attributing the 2017 Bangladesh floods from meteorological and hydrological perspectives, Hydrol. Earth Syst. Sci., 23, 1409-1429, 10.5194/hess-23-1409-2019, 2019.

Tiwari, P., and Joshi, B.: Wildlife in the Himalayan foothills: Conservation and Management, Indus Publishing, 1997.

Uhe, P. F., Mitchell, D. M., Bates, P. D., Sampson, C. C., Smith, A. M., and Islam, A. S.: Enhanced flood risk with 1.5 °C global warming in the Ganges–Brahmaputra–Meghna basin, Environmental Research Letters, 14, 074031, 10.1088/1748-9326/ab10ee, 2019.

---

## Author Comment (AC2) · 3 Oct 2019

**Response to the comments of Reviewer 2**

We are very much thankful to anonymous Reviewer 2 for his time to review the article and provide valuable comments. These comments are useful to improve the overall quality of the article. We will address these in the revised manuscript and accordingly our responses to each comment is given below. We marked our replies in a blue font, while original reviewer comments are presented in a black font.

**Major:**

*RC1: I feel this paper spent a lot of efforts on "meteorological" drivers (which is good and comprehensive). However, to complete its "hydrometeorological drivers" claim in the title, some soil moisture analyses/discussions may be necessary. For example, looking at Fig. 7: it seems to me that the rapid rise of water level for the 2017 floods (different from 1998) may be due to the high antecedent soil moisture in 2017, which is resulted from the very wet spring (April) in 2017. This may have led to the rapid rise of water in 2017. By contrast, in 1998, July and August rainfall anomalies were higher but the prior spring seems to be dry (especially in the lower Brahmaputra), which leaves large room for soil storage and thus leading to much slower flood water rise. I suggest the authors to present some soil moisture analyses, which can help complete their story.*

*• Authors are suggested to reference a very similar analysis conducted in the US (https://journals.ametsoc.org/doi/full/10.1175/JHM-D-18-0038.1), where the researchers found the unique response of a flood was due to similar reasons as found in your study. But they also have provided more detailed analyses on antecedent soil moisture as well as flood celerity in the tributaries above the flooded location (which is also similar to this study). Authors are suggested to gather some soil moisture data (from remote sensing) to perform some analyses to improve their story-telling. Or at the very least, these important "hydro" drivers need to be carefully discussed by the paper.*

**Response:**

AC1: According to reviewer suggestion, we are going to include a short analysis on the soil moisture using NCEP–National Center for Atmospheric research (NCAR) reanalysis volumetric soil moisture product to investigate the soil moisture condition during the flood events. Reanalysis soil moisture is model generated data where atmospheric forcing is used to simulate land surface variability processes (Lu et al., 2005). The NCEP-NCAR uses state of the art analysis for the data assimilation to produce reanalysis data (Kalnay et al., 1996). NCEP reanalysis provides soil moisture in two vertical soil layers at depths 0-10 cm and 10-200 cm. We are using soil moisture at the top layer 0-10 cm to complete the present study. To investigate the surface soil moisture condition over the basin, we calculate soil moisture anomaly before flooding event and at the starting of the event. Four dates have been selected for the two flooding events during the 2017 floods- two dates are selected before the flooding event and

two dates are selected at the starting of the flooding event. We also calculate long-term average soil moisture time series to study the evolution of the moisture in the basin during the monsoon period.

From our new analysis, we found that usually, the soil moisture starts to develop gradually with the contribution of the pre-monsoon rainfall during March to May and reach the maximum level with the start of the monsoon rainfall in the basin (see new Figure RC3a below, showing the climatology over 30 years). The average soil moisture at the topsoil layer of the basin is almost homogeneous over the basin during the monsoon period (see new Figure RC3b below, showing the climatology over 30 years). The homogenous signal in the average over 30 years suggests that this is a regular development during the south Asian monsoon. Figure RC3 c-f provides an outlook of the time series of soil moisture anomaly for the two floods events of 2017. During the first flood event in July, the soil moisture was normal both prior to the rainfall event and at the beginning of the flooding (Figure RC3c and d). Similarly, during the August event the soil moisture was normal prior to the rainfall event (Figure 1e). On the other hand, at the start of the flood event on 11 August a band of positive anomaly of soil moisture was developed in the lower part of the basin (Figure RC3f). The rapid rise of water level was recorded on 12, 13, 14 August. The high soil moisture which developed during the August flood event in the lower part of the basin was associated with the heavy rainfall event which might have contributed to the rapid rise of water level in the river. The amount of rainfall and its spatial distribution contributed to the higher soil moisture in August compared to the event  July.

[Figure]

Figure RC3: (a) Development of soil moisture averaged over the basin between 1987 to 2016; (b) Spatial distribution of average daily volumetric soil-moisture (m³m⁻³) (0-10 cm) during monsoon period (June to September) over 1987-2016; Soil-moisture anomaly time series (c) 01 July 2017; (d) 07 July 2017; (e) 7 August; (f) 11 August 2017 (Data source: NCAR/NCEP reanalysis)

Minor comments:

Minor:
• Fig. 2: I found several arrows do not have any associated texts and were placed in wrong place maybe. Please revise the figure to make sure arrows are correctly drawn.

Response: The small arrows indicates the direction of flow. We will write in the caption that small arrows indicate the direction of flow.

• Fig. 3: why choosing the rain gauge at Syedpur but not others?

Response: We provided basin average rainfall in Figure 3 (a). To show the rainfall events captured by the rainfall gauge, we provided Figure 3(b). Moreover, the rainfall distribution shows heavy rainfall occurred lower sub-basins near international border. Syedpur is located towards the north of the country where heavy rainfall occurred during the 2017 flood event. Therefore, it has been presented as a representative gauge to capture the rainfall event.

• Fig. 4 caption: change to "over the Indian monsoon core zone (rectangular box)";
Response: Caption has been changed

• Fig. 5 caption: missing a bracket;
Response: closing bracket is provided.

• Fig. 8: the starting point of the calculation starts from June. Is it possible to start from spring (e.g. April)? The reason is because the spring rainfall anomaly is important for understanding the antecedent basin wetness condition before flooding (recall my Major Comment earlier).
Response: The starting point of the calculation starts from June. We will include calculation from April in Figure 8.

• P1L19: change "but" to "and"; no transition needed here;
Response: Changed accordingly.

• P1L22: the sentence "Water level and river flow time series …" should be better placed in L28 before "The wavelet analysis";
Response: Changed accordingly.

• P3L2: change "the river flow" to "the water level"
Response: In the second line we talked about the river flow, not water level.

• P3L5: change "consider" to "analyze"
Response: Changed accordingly.
• P6L9: change to "study of Zhang et al. (2017)"
Response: Changed accordingly.

• English presentation of this paper contains several repetitive phrases. For example, (1)

P6L2 and P6L3; can use "it" to replace "the Daubechies wavelet function"; (2) P6L13 and

P6L24: suggest authors to ask help from native speakers to improve English

presentation; here only limited examples are provided but more places need to be thoroughly revised;

Response: In the process of revision, we will look it.

• P6L32: no need to spell out "GEV" again;
Response: Changed accordingly.

• P8L1: change "shifting" to "shift"
Response: Changed accordingly.

• The English presentation of this paper can be much better simplified. For example, P8L23 can be changed to "An El Nino (La Nina) state is defined when ONI exceeds 0.5 degree C (below -0.5 degree C)". Same apply to P7L14;

Response: Changed accordingly.

• P8L28: many of these abbreviations have been defined before, no need to define again. Please check throughout the paper, there are many such cases;
Response: Changed accordingly.

• P9L9: remove "anomalies"! It appeared again later and this use is incorrect;
Response: Changed accordingly.

• P16L9: MJO again; no need to spell out

Response: Changed accordingly.

**References**

Kalnay, E., Kanamitsu, M., Kistler, R., Collins, W., Deaven, D., Gandin, L., Iredell, M., Saha, S., White, G., Woollen, J., Zhu, Y., Chelliah, M., Ebisuzaki, W., Higgins, W., Janowiak, J., Mo, K. C., Ropelewski, C., Wang, J., Leetmaa, A., Reynolds, R., Jenne, R., and Joseph, D.: The NCEP/NCAR 40-Year Reanalysis Project, Bulletin of the American Meteorological Society, 77, 437-472, 10.1175/1520-0477(1996)077<0437:tnyrp>2.0.co;2, 1996.

Lu, C.-H., Kanamitsu, M., Roads, J. O., Ebisuzaki, W., Mitchell, K. E., and Lohmann, D.: Evaluation of Soil Moisture in the NCEP–NCAR and NCEP–DOE Global Reanalyses, Journal of Hydrometeorology, 6, 391-408, https://doi.org/10.1175/JHM427.1, 2005.